# BMAttn: Block-Aligned Mixed-Precision Attention Quantization for LLM Inference

## Abstract

The proliferation of Large Language Models (LLMs) with extended context windows is severely hampered by the quadratic complexity of the self-attention mechanism. Existing acceleration methods, such as sparse attention and quantization, often employ uniform compression strategies that are misaligned with the non-uniform distribution of information importance within attention maps. This leads to a suboptimal trade-off between computational efficiency and model accuracy. To address this, we introduce Block-based Mixed-precision Attention (BMAttn), a novel framework that enables fine-grained, importance-aware precision while maintaining a hardware-friendly structure. BMAttn partitions each attention head into high-precision, low-precision, and sparse regions. To ensure computational regularity, these regions are block-aligned. To adapt to varying input lengths, their boundaries are dynamically adjusted using a lightweight affine windowing mechanism. We further propose a saliency-weighted calibration method and a layer-adaptive regularizer to automatically determine the optimal parameters, achieving a superior accuracy-efficiency balance. BMAttn achieves a speedup of up to 3.3× without any accuracy degradation, and a 5× speedup with only a 1% accuracy loss.

## 1 Introduction

Recent advancements in Large Language Models (LLMs) have yielded remarkable capabilities across a wide range of applications (Liu et al., 2024; Achiam et al., 2023; Yang et al., 2023). A prominent trend in LLM development is the expansion of context windows, with models like Gemini (Comanici et al., 2025) and Kimi (Team et al., 2025) now capable of processing sequences of up to one million tokens. However, this expansion is fundamentally constrained by the underlying self-attention mechanism (Vaswani et al., 2017). Its computational complexity scales quadratically ($O(L^2)$) with the input sequence length $L$, posing a significant bottleneck to the efficiency, scalability, and practical deployment of these powerful models.

Existing methods to mitigate this cost, such as sparse attention (Zhang et al., 2025b; Fu et al., 2024; Gao et al., 2024; Xu et al., 2025; Xiao et al., 2024b) and quantization (Zhang et al., 2024b;a), often rely on a uniform compression strategy. Sparse attention, for instance, is confined to a coarse, binary 'keep-or-discard' decision, while typical quantization schemes apply a homogeneous bit-width across all connections. These approaches are fundamentally misaligned with the non-uniform nature of attention. As observed in many models, attention heads exhibit heterogeneous patterns: some specialize in local, fine-grained interactions requiring high precision, while others capture global, contextual cues that can be represented more coarsely. A uniform strategy fails to adapt the computational effort to this variance in information importance, leading to a suboptimal trade-off between efficiency and accuracy.

This reveals a critical need for a framework that can process information with fine-grained, importance-aware precision. The central challenge, however, lies in implementing such adaptivity without incurring the performance penalties of irregular memory access and computation, which would nullify any theoretical gains on modern hardware. To resolve this tension, we introduce **Block-Aligned Mixed-precision Attention (BMAttn)**, a novel framework that co-designs a dynamic precision allocation scheme with a hardware-friendly computational structure. The core principle of BMAttn is to partition the attention computation for each head into three regions: a high-precision zone for critical short-range interactions, a low-precision zone for less critical long-range context,

and a sparse zone where negligible connections are pruned entirely. To ensure hardware efficiency, this partitioning is not applied at the token level but is instead **block-aligned**: the sequence is divided into fixed-size blocks, and precision is assigned at this coarser granularity. This blockwise structure creates a regular, staircase-shaped computational layout that is highly amenable to GPU acceleration and compatible with optimized kernels like FlashAttention Dao (2023).

To enable dynamic adaptation to varying context lengths, we introduce an **affine windowing mechanism**. Instead of using fixed boundaries, the size of the high- and low-precision regions for each head is parameterized by a simple affine function of the context length. This formulation allows each head to dynamically and flexibly adjust its attention span—for instance, growing its high-precision zone proportionally with the sequence length—while maintaining a regular block structure. This elegantly reduces the complex problem of per-token precision assignment to learning a small set of affine parameters for each head, unifying fine-grained adaptivity with structured efficiency. Finally, to make this framework practically effective, we propose a calibration method to automatically determine the optimal affine parameters. Recognizing that standard metrics like MSE are insufficient, we develop a **saliency-weighted calibration process** that aligns parameter tuning with the true perceptual importance of attention scores. Furthermore, we observe that model layers exhibit heterogeneous sensitivity to compression. We address this with a **layer-adaptive retention regularizer**, which enforces stricter precision targets on fragile shallow layers while allowing for more aggressive compression in robust deeper ones. This layered approach preserves model integrity while maximizing efficiency gains.

In summary, our contributions are as follows:

- We introduce BMAttn, an attention framework that unifies a mixed-bit representation with a block-aligned structure and an affine windowing mechanism to achieve both fine-grained adaptivity and hardware-friendly execution.

- We propose a sophisticated calibration method featuring a saliency-weighted metric for accurate parameter setting and a layer-adaptive retention regularizer to achieve a superior accuracy-efficiency trade-off.

- We conduct extensive experiments demonstrating that BMAttn achieves up to a 3.3x speedup on long-context language modeling tasks without any degradation in accuracy.

## 2 RELATED WORKS

**Algorithmic Optimizations for Efficient Attention.** Algorithmic approaches primarily focus on mitigating the quadratic complexity of the attention mechanism. Early works introduced structured sparsity through fixed patterns, such as the local windows in Swin Transformer (Liu et al., 2021) and Twins (Chu et al., 2021). To better handle long-sequence tasks, subsequent methods adopted more adaptive context management. For instance, StreamingLLM (Xiao et al., 2023), InfLLM (Xiao et al., 2024a), and LongLoRA (Chen et al., 2023) selectively retain or expand the context to preserve crucial information. More recently, a line of inference-centric designs has emerged, including Minference (Jiang et al., 2024), SkipAttention(Venkataramanan et al., 2023), and SpargeAttention Zhang et al. (2025b). These methods dynamically identify and skip near-zero attention entries during generation, achieving multi-fold speedups without requiring retraining. Further extensions like XAttention (block pruning) (Xu et al., 2025) and DuoAttention (head-level cache partitioning) (Xiao et al., 2024b) demonstrate that structured sparsity can simultaneously reduce computation and memory demands while maintaining model accuracy. A complementary line of algorithmic work targets the KV cache, which is a major bottleneck in autoregressive decoding due to its memory bandwidth demands. By distinguishing retrieval-critical heads from local ones, or by compressing and offloading cache states, methods like ShadowKV (Sun et al., 2024) can reduce GPU memory usage by 2–6× and boost throughput on million-token contexts by up to 3×. The plug-and-play nature of these techniques makes them highly suitable for real-world inference scenarios.

**System-Level and Hardware-Aware Optimizations.** Orthogonal to algorithmic modifications, system-level optimizations aim to maximize hardware utilization through efficient kernel implementations. The xFormers library (Lefaudeux et al., 2022) provides a collection of modular, high-performance kernels. A pivotal contribution in this area is FlashAttention (Dao et al., 2022), which introduced I/O-aware tiling to minimize the high cost of GPU HBM access, a technique later refined in FlashAttention-2 (Dao, 2023) and FlashAttention-3 (Shah et al., 2024). In parallel, quantization

offers another path to acceleration. I-BERT (Kim et al., 2021) first showed the feasibility of INT8 quantization for RoBERTa-like models. More recently, SageAttention (Zhang et al., 2024b;a; 2025a) generalized this by demonstrating that a full INT8 attention implementation can outperform FP16 FlashAttention variants in both speed and accuracy, functioning as a drop-in replacement.

In summary, the pursuit of efficient Transformers has spurred innovations across three complementary fronts: algorithmic sparsification, KV cache management, and low-level kernel optimization. These approaches are largely orthogonal, and their combination can yield additive performance gains.

## 3 PRELIMINARIES

**Full attention.** Self-attention projects inputs into queries, keys, and values $\mathbf{Q}, \mathbf{K}, \mathbf{V} \in \mathbb{R}^{L \times d_k}$, computes scores $\mathbf{S} = \mathbf{Q}\mathbf{K}^\top$, compute attention weight $\mathbf{W} = \mathrm{softmax}(\mathbf{S}/\sqrt{d_k})$, and outputs $\mathbf{O} = \mathbf{W}\mathbf{V}$. The score computation scales as $\mathcal{O}(L^2 d_k)$ and dominates for long sequences.

**Windowed attention.** Attention computation is reduced by restricting each query $q_i$ to a subset of keys $\mathcal{J}_i \subset \{0, \ldots, L-1\}$ via masking: $\mathbf{S}'_{ij} = \mathbf{S}_{ij}$ if $j \in \mathcal{J}_i$ and $\mathbf{S}'_{ij} = -\infty$ otherwise. A common design uses a fixed attention sink of length $s$ and a local window of size $k$: $\mathcal{J}_i = \{0, \ldots, s-1\} \cup \{j \mid \max(s, i-k+1) \leq j \leq i\}$. This reduces computed scores from $L^2$ to roughly $L(s+k)$, yielding linear complexity in $L$.

**Quantization.** Quantization accelerates matrix multiplication (e.g., $\mathbf{C} = \mathbf{A}\mathbf{B}$) by mapping high-precision matrices to low-precision integers via a quantizer $\psi$ and de-quantizer $\psi^{-1}$. With $(\delta_A, \hat{\mathbf{A}}) = \psi(\mathbf{A})$ and $(\delta_B, \hat{\mathbf{B}}) = \psi(\mathbf{B})$, we compute $\hat{\mathbf{C}} = \hat{\mathbf{A}}\hat{\mathbf{B}}$ using integer arithmetic and recover $\mathbf{C} \approx \psi^{-1}_{\delta_A, \delta_B}(\hat{\mathbf{C}}) = \hat{\mathbf{C}} \cdot \delta_A \delta_B$. Quantization granularity controls which elements share a scale $\delta$; for symmetric INT8, $\delta_{\mathrm{group}} = \max(|\mathbf{X}_{\mathrm{group}}|)/127$. Beyond per-tensor scaling, per-block quantization partitions matrices into blocks and assigns a scale per block, improving fidelity to local statistics with modest overhead.

For Attention, quantization acts on the matrix-multiplication operands: (i) $\mathbf{Q}\mathbf{K}^\top$ path via quantized $\mathbf{Q}$ and $\mathbf{K}$; (ii) $\mathbf{W}\mathbf{V}$ path via quantized $\mathbf{V}$ and quantized attention weights $\mathbf{W}$. Attention quantization enables faster computation and memory access.

## 4 METHOD

We propose **BMAttn**, a **B**lock-aligned **M**ixed-precision **A**ttention framework for efficient LLM inference. BMAttn allocates precision heterogeneously across the attention map while maintaining a regular compute pattern compatible with GPU kernels such as FlashAttention (Dao, 2023). It uses high precision for short-range dependencies, low precision for mid- to long-range interactions, and sparsity (0-bit) for negligible links. Precision boundaries adapt with sequence length via simple affine functions and are calibrated offline using saliency-weighted metrics and a layer-adaptive retention regularizer.

### 4.1 BLOCK-ALIGNED MIXED-PRECISION ATTENTION

**Motivation.** Scaling the context window of LLMs is fundamentally constrained by the $O(L^2)$ cost of self-attention in both computation and memory movement. At the same time, two consistent empirical observations make *uniform* precision and sparsity assignments intrinsically suboptimal: (i) *Distance heterogeneity*: attention mass and its semantic criticality decay with causal distance. Short-range links encode fine-grained lexical/syntactic cues, whereas mid/long-range links contribute broader discourse-level signals. (ii) *Head heterogeneity*: different heads specialize in distinct ranges and patterns (local vs. global). These factors imply that a one-size-fits-all policy (e.g., a single sparsity pattern or a single bit-width across the entire map) either wastes compute on unimportant regions or harms accuracy by underrepresenting crucial ones.

However, simply making precision adaptive at a fine granularity is not sufficient: *naive* token-level mixed precision or unaligned per-row cutoffs introduce irregular memory access, warp divergence, and scattered reads/writes that nullify theoretical savings on modern GPUs. Similarly, fixed, hand-

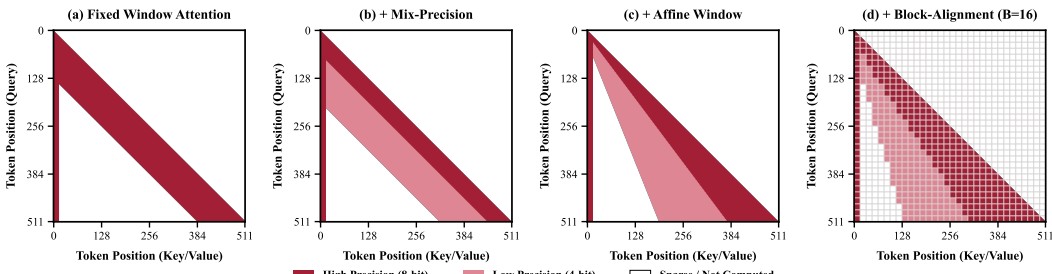

Figure 1: Comparison of different attention window schemes. (a) Window Attention: Fixed-size sliding window. (b) Constant Precision Tiers: High and low precision regions. (c) Affine Window: Adaptive precision boundaries based on context length. (d) Block-Alignment: Attention is divided into contiguous blocks, optimizing hardware efficiency.

tuned distance cutoffs fail to generalize across varying context lengths; a window suitable at $L = 4\text{K}$ can be over- or under-aggressive at $L = 128\text{K}$.

**Distance-indexed Three-zone Decomposition.** As shown in Figure 1, we partition each head's attention into three monotone bands along the causal distance: a *high-precision* (HP) zone for short-range, high-saliency links; a *low-precision* (LP) zone for mid/long-range links that tolerate lower bit-width; and a *sparse* (SP) zone where negligible links are skipped entirely. This tri-partition balances accuracy (HP), efficiency via cheaper arithmetic (LP), and outright pruning (SP). It also naturally accommodates *sink tokens* (Xiao et al., 2023), which we place at the sequence front and always keep in HP to stabilize retrieval over extremely long contexts.

**Zone Decomposition by Affine Window.** Let $L_{\text{seq}}$ be the full sequence length, $L_{\text{sink}}$ the number of fixed-position sink tokens at the sequence front, and $L_{\text{ctx}} = L_{\text{seq}} - L_{\text{sink}}$ the non-sink context length. Under strict causality, the query–key distance is $d_{i,j} = i - j \geq 0$. For each head $h$, we define two affine thresholds that control the extent of the high-precision (HP) and low-precision (LP) regions:

$$d_{\text{hp}}^{(h)} = w_{\text{hp}}^{(h)} \cdot L_{\text{ctx}} + b_{\text{hp}}^{(h)}, \qquad d_{\text{lp}}^{(h)} = w_{\text{lp}}^{(h)} \cdot L_{\text{ctx}} + b_{\text{lp}}^{(h)}, \tag{1}$$

with the ordering constraint $0 \leq d_{\text{hp}}^{(h)} \leq d_{\text{lp}}^{(h)} \leq L_{\text{ctx}}$. The scaling coefficients $w_{\{\text{hp,lp}\}}^{(h)}$ govern proportional growth with $L_{\text{ctx}}$, while the biases $b_{\{\text{hp,lp}\}}^{(h)}$ provide head-specific offsets. These parameters are *fixed per head* after offline calibration and deterministically produce context-aware thresholds at inference for any $L_{\text{ctx}}$. The thresholds in Eqs. equation 1 partition the causal attention into three disjoint regions by distance: (i) HP: $0 \leq d_{i,j} \leq d_{\text{hp}}^{(h)}$; (ii) LP: $d_{\text{hp}}^{(h)} < d_{i,j} \leq d_{\text{lp}}^{(h)}$; (iii) Sparse: $d_{i,j} > d_{\text{lp}}^{(h)}$. Sink tokens reside at the first $L_{\text{sink}}$ key positions and are always treated as HP for any query, independent of their distances; they are excluded from the dynamic partitioning governed by $d_{\text{hp}}^{(h)}$ and $d_{\text{lp}}^{(h)}$.

**Block-Aligned Mixed-Precision Window.** To realize the affine windows efficiently on modern GPUs, we convert the continuous thresholds into *block-aligned* boundaries and execute attention in tiles that match hardware-friendly memory layouts (FlashAttention-compatible). Let $B$ be the block size. We snap the affine thresholds to the nearest lower multiple of $B$:

$$D_{\text{hp}}^{(h)} = \lfloor d_{\text{hp}}^{(h)} \rfloor_B, \qquad D_{\text{lp}}^{(h)} = \lfloor d_{\text{lp}}^{(h)} \rfloor_B, \tag{2}$$

where $\lfloor \cdot \rfloor_B$ denotes *floor-to-multiple-of-$B$* rounding. The resulting zone definition is:

$$\mathcal{Z}_{\text{hp}}^{(h)} = \{(i,j) \mid j < i, \text{ sink}(j) \text{ or } 0 \leq d_{i,j} \leq D_{\text{hp}}^{(h)}\}, \tag{3}$$

$$\mathcal{Z}_{\text{lp}}^{(h)} = \{(i,j) \mid j < i, \ d_{\text{hp}}^{(h)} < d_{i,j} \leq D_{\text{lp}}^{(h)}\}, \tag{4}$$

$$\mathcal{Z}_{\text{sp}}^{(h)} = \{(i,j) \mid j < i, \ d_{i,j} > D_{\text{lp}}^{(h)}\}, \tag{5}$$

and the per-row pattern is staircase-shaped because $d_{i,j} = i - j$ increases monotonically along the causal axis. We adopt mixed precision at the tile granularity: HP: 8-bit quantization for Q/K/V

Table 1: Comparison of performance metrics during pruning of long-range attention.

| Remove Ratio | Cos Sim | RMSE | Relative L1 | ASC | **RDW** | **IPW** | Ppl. (FP=7.458) |
|---|---|---|---|---|---|---|---|
| 95% | 99.92% | 0.1186 | 0.0887 | 0.129 | 0.9714 | 0.9123 | **44.3892** |
| 90% | 99.95% | 0.0881 | 0.0616 | 0.103 | 0.9223 | 0.8880 | **21.4239** |
| 80% | 99.98% | 0.0594 | 0.0368 | 0.087 | 0.8378 | 0.7955 | **16.9872** |
| 70% | 99.99% | 0.0303 | 0.0181 | 0.056 | 0.6945 | 0.6131 | **14.2538** |

tensors and for post-softmax attention weights used in value aggregation. LP: 4-bit quantization for Q/K/V tensors and for post-softmax attention weights. SP: 0-bit (skipped). Because computation in each tile has the same precision, we can efficiently use the tile-by-tile computation pattern of FlashAttention (Dao, 2023) without mask.

## 4.2 SALIENCY-WEIGHTED CALIBRATION

We calibrate the per-head affine parameters $\Theta = \{(w_{\text{hp}}^{(h)}, b_{\text{hp}}^{(h)}, w_{\text{lp}}^{(h)}, b_{\text{lp}}^{(h)})\}$ offline on a small calibration set. The objective is to choose the *smallest* windows (i.e., most compressed boundaries) that satisfy saliency-retention constraints defined below.

### 4.2.1 SALIENCY-WEIGHTED METRIC

**Motivation.** Standard reconstruction metrics (e.g., MSE, RMSE, cosine similarity) and attention coverage are dominated by short-range pairs and thus are insensitive to errors that arise from pruning or over-compressing rare yet semantically crucial long-range links. Under strict causal masking, the number of query–key pairs at distance $d$ is proportional to $L - d$ for a sequence of length $L$, so small $d$ overwhelmingly contributes to any unweighted aggregate. As a result, large changes to a small set of long-range entries barely shift the global metric even when they substantially affect model behavior. As shown in Table 1, conventional metrics like MSE, Cosine Similarity, RMSE, and attention score coverage (ASC) remain mostly unchanged even as model performance deteriorates significantly. This shows that these metrics overlook critical long-range dependencies. To counter this strong locality bias, we should reweight attention statistics by a distance-dependent saliency, so that calibration criteria reflect the perceptual importance of long-range dependencies.

**Saliency construction.** For a given layer $l$ and head $h$, let $\mathbf{W}^{(l,h)} \in \mathbb{R}^{L \times L}$ be the attention weight under causal masking, and let $d_{i,j} = i - j \geq 0$. We define a saliency matrix

$$S_{i,j}^{(l,h)} = \varphi^{(l,h)}(d_{i,j}) \cdot \mathbf{W}_{i,j}^{(l,h)}, \tag{6}$$

where $\varphi^{(l,h)}(d)$ compensates for the empirical rarity of long-range pairs. We design two forms:

$$\textbf{RDW (Relative-Distance Weighting):} \quad \varphi_{\text{RDW}}^{(l,h)}(d) = \frac{d}{L_{\text{ctx}}}, \tag{7}$$

$$\textbf{IPW (Inverse-Propensity Weighting):} \quad \varphi_{\text{IPW}}^{(l,h)}(d) = \frac{L_{\text{ctx}}}{p^{(l,h)}(d) + \epsilon}. \tag{8}$$

As shown in Figure 2, $p^{(l,h)}(d)$ is the attention weight distribution estimated on the calibration set with distance bucketing [1], and $\epsilon > 0$ is a small constant for numerical stability. RDW is a lightweight heuristic; IPW is a importance-sampling correction.

### 4.2.2 LAYER-ADAPTIVE RETENTION REGULARIZER

**Motivation.** While a saliency-based metric captures attention importance at the head level, it overlooks the heterogeneous sensitivity of layers. Prior work shows that shallow layers are more fragile under pruning and quantization, as they preserve low-level token information, whereas deeper layers are more robust due to redundancy in higher-level representations (Huang et al., 2025). This

---

[1]See Appendix H for details to compute attention weight distribution across distances.

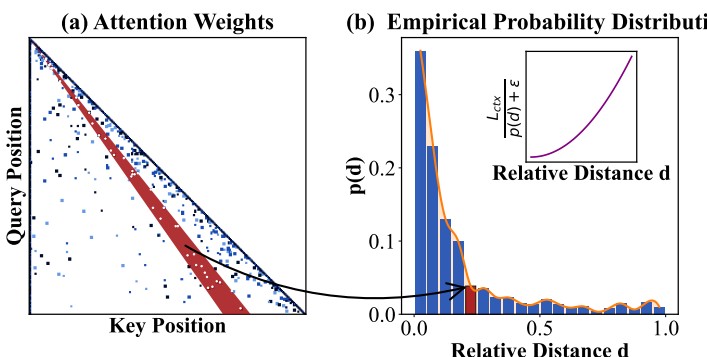

Figure 2: Attention weight and its attention weight distribution.

asymmetry makes uniform retention targets suboptimal. We therefore introduce a layer-adaptive retention regularizer.

**Method.** We design the regularizer as a depth-dependent constraint on saliency energy. Instead of defining the schedule via absolute start and end points, we re-parameterize it using two more intuitive hyperparameters: a global average retention target $\mu_\Gamma \in (0, 1]$ and a linear decay factor $\delta \geq 0$. This provides direct control over the overall compression level and its distribution across layers.

Formally, let $L_{total}$ be the total number of layers in the model, indexed from $l = 0$ to $L_{total} - 1$. The layer-specific retention target $\Gamma_l$ is defined as:

$$\Gamma_l = \mu_\Gamma + \delta \left( \frac{L_{total} - 1}{2} - l \right). \tag{9}$$

This formulation ensures that the average retention across all layers is exactly $\mu_\Gamma$, since the term $\left( \frac{L_{total}-1}{2} - l \right)$ has a mean of zero over $l \in [0, L_{total} - 1]$. The decay factor $\delta$ controls the slope of the linear schedule: a positive $\delta$ assigns a higher-than-average retention target to shallow layers (where $l < (L_{total} - 1)/2$) and a lower-than-average target to deep layers (where $l > (L_{total} - 1)/2$), thus enforcing our desired conservative-to-aggressive policy.

### 4.2.3 CONSTRAINED CALIBRATION ALGORITHM

Having defined the saliency metric and the layer-adaptive retention regularizer, we now describe how the affine window parameters are calibrated. Let $E_{l,h}$ denote the total saliency energy of head $h$ in layer $l$, aggregated over calibration data:

$$E_{l,h} = \sum_{i,j} S_{i,j}^{(l,h)}. \tag{10}$$

For each head $(l, h)$, the affine window parameters $(w, b)$ must satisfy the constraint imposed by the retention schedule:

$$\sum_{d \leq d_{lp}^{(h)}} S_{i,j}^{(l,h)} \geq \Gamma_l \cdot E_{l,h}, \tag{11}$$

and analogously for the high-precision zone with a more strict target $\gamma \cdot \Gamma_l \cdot E_{l,h}$, where $0 < \gamma < 1$. The goal is to determine the optimal per-head parameters $\Theta = \left\{ (w_{hp}^{(h)}, b_{hp}^{(h)}, w_{lp}^{(h)}, b_{lp}^{(h)}) \mid \forall(l, h) \right\}$ for all heads $(l, h)$ in the model.

For each head, we formulate a constrained optimization problem:

$$\min_{w_{hp}, b_{hp}, w_{lp}, b_{lp}} \quad \mathbb{E}_{\mathcal{S}} \left[ d_{hp}^{(h)} \right] \; + \; \mathbb{E}_{\mathcal{S}} \left[ d_{lp}^{(h)} \right] \tag{12}$$

$$\text{s.t.} \quad \sum_{d \leq d_{lp}^{(h)}} S_{i,j}^{(l,h)} \; \geq \; \Gamma_l \cdot E_{l,h}, \qquad \sum_{d \leq d_{hp}^{(h)}} S_{i,j}^{(l,h)} \; \geq \; \gamma \cdot \Gamma_l \cdot E_{l,h}, \tag{13}$$

where $\gamma$ is a hyperparameter controlling the span of HP vs. LP, and $\Gamma_l$ is the retention target imposed by the layer-adaptive retention regularizer. The detailed offline calibration algorithm is shown in Appendix A.

# 5 EXPERIMENTS

## 5.1 SETUP

**Models.** We evaluate BMAttn across three representative large language models (LLMs): Qwen2.5 (7B) (Qwen et al., 2025), Llama3 (8B) (Dubey et al., 2024), and GLM4 (9B) (GLM et al., 2024). These models span different architectures and training regimes, providing a diverse testbed for assessing the performance and efficiency of our method.

**Datasets and Metrics.** Our evaluation spans both standard and long-context benchmarks: WikiText-2 (Merity et al., 2016) for language modeling perplexity, MMLU (Hendrycks et al., 2020) for multi-task understanding, LongBench (Bai et al., 2023) for long-context reasoning, and RULER (Hsieh et al., 2024) for reasoning and reading comprehension. Metrics include perplexity (lower is better), accuracy, and task-specific scores, depending on the nature of the task.

**Baselines.** We compare BMAttn with several state-of-the-art attention mechanisms, including FlashAttention-2 (Dao, 2023), SageAttention (Zhang et al., 2024b), and SageAttention2 (Zhang et al., 2024a). These baselines are widely regarded as strong implementations in the efficient attention space and provide a robust comparison across both accuracy and efficiency.

**Implementation Details.** For quantization, we employ a hybrid scheme where $Q$, $K$, and $W$ are quantized per block, while $V$ is quantized per channel. This approach ensures efficient block-aligned execution while preserving accuracy in the output representations. In all experiments, we set the average retention rate $\mu_\Gamma = 0.8$, decay factor $\delta = 0.01$, and $\gamma = 0.9$. For metric, we mainly report the results using IPW. We evaluated our model on a hardware device featuring a 1 Tbps memory bandwidth. This device boasts a computation capacity of 83 TFLOPs (16-bit precision), alongside 660.6 TOPS 8-bit precision) and 1321.2 TOPS (4-bit precision). Additionally, the code was developed based on the open-source implementation of SageAttention2.

## 5.2 ACCURACY RESULT

Table 2: Accuracy and Efficiency Comparison of BMAttn with Baseline Methods.

| Method | WikiText (Ppl.)↓ | MMLU (Acc.)↑ | LongBench↑ | RULER↑ | Avg. Bits | Speedup |
|---|---|---|---|---|---|---|
| Qwen2.5-7B-Instruct (Qwen et al., 2025) | | | | | | |
| FlashAttention-2 (Dao, 2023) | 7.458 | 0.717 | 52.58 | 94.05 | 16 | 1.00× |
| SageAttention-8b (Zhang et al., 2024b) | 7.463 | 0.716 | 52.69 | 93.59 | 8 | 2.00× |
| SageAttention2-4b (Zhang et al., 2024a) | 7.582 | 0.702 | 51.70 | 88.67 | 4 | 2.93× |
| BMAttn | 7.461 | 0.716 | 52.67 | 94.01 | 5.75 | 3.26× |
| Llama3.1-8B-Instruct (Dubey et al., 2024) | | | | | | |
| FlashAttention-2 (Dao, 2023) | 7.217 | 0.629 | 54.09 | 91.27 | 16 | 1.00× |
| SageAttention-8b (Zhang et al., 2024b) | 7.223 | 0.627 | 54.07 | 90.00 | 8 | 2.00× |
| SageAttention2-4b (Zhang et al., 2024a) | 7.461 | 0.598 | 52.89 | 85.24 | 4 | 2.93× |
| BMAttn | 7.220 | 0.627 | 54.00 | 91.15 | 5.99 | 3.11× |
| GLM-4-9B-Chat (GLM et al., 2024) | | | | | | |
| FlashAttention-2 (Dao, 2023) | 11.937 | 0.682 | 53.53 | 92.33 | 16 | 1.00× |
| SageAttention-8b (Zhang et al., 2024b) | 11.997 | 0.680 | 53.19 | 92.28 | 8 | 2.00× |
| SageAttention2-4b (Zhang et al., 2024a) | 12.219 | 0.643 | 50.38 | 88.96 | 4 | 2.93× |
| BMAttn | 11.965 | 0.681 | 53.26 | 92.26 | 5.82 | 3.20× |

In Table 2, we present the results comparing BMAttn with the baselines. The findings highlight that BMAttn achieves competitive accuracy with state-of-the-art methods, while delivering substantial improvements in computational efficiency across all the models and datasets. Specifically, for Qwen2.5-7B-Instruct, BMAttn maintains near-lossless accuracy compared to the FlashAttention-

2 and SageAttention-8b, while achieving a 3.26× speedup. In addition, `BMAttn` outperforms SageAttention2-4b, both in terms of accuracy (52.67 vs. 51.70 on LongBench) and speedup (3.26× vs. 2.93×). These results validate the effectiveness of `BMAttn` in balancing high accuracy with significant efficiency gains, demonstrating its potential as a robust solution for large language models.

## 5.3 EFFICIENCY RESULT

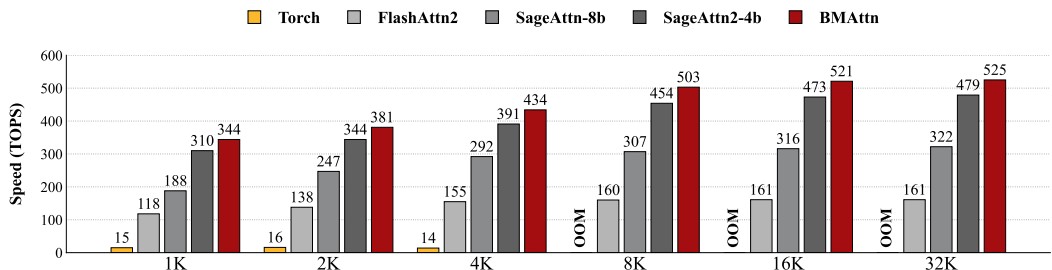

Figure 3: Lossless Efficiency comparison between **BMAttn** and baselines.

**Lossless Efficiency.** We first evaluate efficiency under a lossless setting using `headdim=128` and causal masking (Vaswani et al., 2017). As shown in Fig. 3, `BMAttn` achieves consistent speedups across varying sequence lengths, reaching 1.63× over SageAttention-8b and 1.10× over SageAttention2-4b. Notably, SageAttention2-4b operates at lower precision and incurs accuracy degradation, whereas `BMAttn` preserves near-lossless accuracy. Furthermore, as the sequence length increases, the acceleration factor of `BMAttn` continues to grow, which indicates that `BMAttn` is able to exploit the increased context effectively. This behavior suggests that `BMAttn` becomes increasingly more efficient as the task scales, likely due to its ability to optimize memory access patterns and computational load through block-alignment and precision adaptation.

Table 3: Aggressive efficiency trade-off on LongBench. `BMAttn` achieves higher acceleration factors while sustaining accuracy. Green values indicate performance better than the full-precision (FP) model, while red values indicate performance worse than SageAttn2-4b.

| Metric | $\mu_\Gamma$ $(\delta = 0.01, \gamma = 0.9)$ | | | | | | | | | | | |
|---|---|---|---|---|---|---|---|---|---|---|---|---|
| | 0.90 | 0.85 | 0.80 | 0.75 | 0.70 | 0.65 | 0.60 | 0.55 | 0.50 | 0.45 | 0.40 | 0.35 |
| LongBench | **53.56** | **53.36** | **53.13** | **52.89** | **52.67** | 52.49 | 52.32 | 52.21 | 51.86 | 51.15 | 50.78 | 49.09 |
| Avg. Bits | 7.24 | 6.91 | 6.57 | 6.12 | 5.75 | 5.45 | 5.05 | 4.64 | 4.25 | 3.79 | 3.36 | 2.90 |
| Speedup | 2.69× | 2.82× | 2.96× | 3.12× | 3.26× | 3.54× | 3.81× | 4.14× | 4.53× | 5.01× | 5.63× | 6.45× |

**Aggressive Efficiency.** We further explore aggressive efficiency by progressively lowering retention targets. In particular, the results in Tab. 3 reveal three important insights. First, by progressively reducing the average bitwidth (from 7.24 down to 2.90), `BMAttn` still sustains competitive LongBench scores above 52 under ∼4× acceleration, clearly outperforming SageAttention2-4b (51.70 @ 2.93×). This highlights its robustness in balancing accuracy and efficiency under moderate compression. Second, even in the extreme regime of >6× acceleration, the model maintains scores around 49, which remain usable for many downstream tasks. This demonstrates that `BMAttn` can still provide meaningful outputs when deployed in highly resource-constrained environments. Third, the adaptive calibration mechanism plays a crucial role by ensuring a smooth trade-off: rather than suffering abrupt performance degradation when bitwidth decreases, the model achieves a more flexible and stable efficiency–accuracy balance. These observations confirm that `BMAttn` is not only effective at moderate acceleration but also resilient at extreme compression levels.

## 5.4 ABLATION STUDY

**Affine Window Ablation.** To demonstrate the importance of affine window mechanism, we compare our affine window against a fixed-window baseline, whose window sizes are calibrated

Table 4: Ablation on window design.

| Seqlen | Fixed Window | | Affine Window | |
|---|---|---|---|---|
| | RULER | Speedup | RULER | Speedup |
| 4K | 94.05 | 1.91× | 94.01 | 3.26× |
| 8K | 92.39 | 3.12× | 92.48 | 3.14× |
| 16K | 89.15 | 3.45× | 91.89 | 3.24× |
| 32K | 44.69 | 5.45× | 88.69 | 3.26× |
| 64K | 23.54 | 9.11× | 83.20 | 3.29× |
| 128K | 6.88 | 19.81× | 73.35 | 3.33× |

Table 5: Ablation on SWM and LRR.

| Model | SWM | LRR | LongBench ↑ |
|---|---|---|---|
| Qwen2.5-7B-Instruct | ✗ | ✗ | 46.53 |
| | ✗ | ✓ | 50.06 |
| | ✓ | ✗ | 52.01 |
| | ✓ | ✓ | **52.67** |
| Llama3.1-8B-Instruct | ✗ | ✗ | 48.66 |
| | ✗ | ✓ | 51.38 |
| | ✓ | ✗ | 53.04 |
| | ✓ | ✓ | **54.00** |

at 8k tokens and then held constant. Table 4 reveals the fundamental flaws of this rigid strategy, which is only effective when the inference context closely matches the calibration context. On short sequences (4k), the fixed window is overly conservative; it preserves a large region at high precision that contains little valuable information, leading to suboptimal compression and thus limited speedup (1.91× vs. 3.26×). Conversely, on long sequences (128k), the window becomes severely restrictive. It fails to cover essential long-range dependencies, causing a catastrophic accuracy collapse (RULER score of 6.88). In contrast, our affine window gracefully adapts to all scenarios, maintaining robust accuracy while delivering a consistent ∼3.3× speedup. These results confirm that dynamic affine scaling is essential for building a generalizable and efficient model.

**Components Ablation.** The ablation in Table 5 clearly demonstrates the necessity of both the saliency-weighted metric (SWM) and the layer-adaptive retention regularizer (LRR). Without LRR, the model suffers a severe drop in LongBench score (46.53), confirming that uniform retention targets are ill-suited given the heterogeneous sensitivity of different layers. Introducing LRR alone already restores performance to 50.06, as depth-aware allocation prevents shallow layers from collapsing under aggressive pruning. SWM provides complementary benefits: by reweighting attention calibration toward semantically crucial long-range pairs, it recovers additional accuracy (52.01). When combined, SWM+LRR achieves the best results, reaching 52.67 on Qwen2.5-7B and 54.00 on Llama3.1-8B. This synergy reflects their distinct yet aligned roles—SWM corrects locality bias at the head level, while LRR counteracts asymmetry across depth—together producing a balanced strategy that generalizes across architectures.

**Calibration Cost.** In our method, the calibration process involves offline optimization of affine window parameters for each attention head, which requires a one-time computation. On the Qwen2.5-7B-Instruct model, the peak GPU memory usage during calibration is 14,362 MB, and the total calibration time is 1 minute and 15 seconds. This calibration cost is a one-time expense that occurs during the model setup phase and does not affect the subsequent inference time, which remains efficient due to the optimizations in precision and sparsity handling during the attention computation. This calibration time and memory usage are reasonable given the significant improvements in computational efficiency and accuracy that `BMAttn` delivers during inference.

## 6 CONCLUSION

In this work, we presented `BMAttn`, a block-aligned mixed-precision attention framework that reconciles the tension between fine-grained adaptivity and hardware efficiency. By decomposing attention into high-, low-, and zero-precision zones, and aligning them with blockwise affine windows, `BMAttn` achieves structured efficiency while preserving critical dependencies. Our saliency-weighted calibration and layer-adaptive regularization further ensure that compression decisions align with both semantic importance and layer sensitivity. Extensive experiments across WikiText, MMLU, LongBench, and RULER demonstrate that `BMAttn` consistently outperforms state-of-the-art baselines, delivering up to 3.3× acceleration without accuracy loss and sustaining usable performance even at > 6× compression. Future work may explore integrating `BMAttn` with complementary compression strategies (e.g., KV-cache quantization, structured sparsity, low-rank adaptation), extending adaptive calibration to dynamic online settings, and co-designing mixed-precision schedules with hardware-aware compilers.

ETHICS STATEMENT

This work focuses on improving the efficiency of large language models through the `BMAttn` mixed-precision attention mechanism. No human subjects, sensitive data, or privacy issues were involved in the development or testing of the model. Additionally, the methodology does not introduce any new ethical concerns regarding discrimination, bias, or harmful insights. The application of `BMAttn` is intended to enhance model performance and efficiency, with no intention to promote malicious or unethical use. We have adhered to standard research integrity practices, ensuring transparency and fairness throughout the development of this method.

REPRODUCIBILITY STATEMENT

To ensure the reproducibility of our findings, detailed implementation instructions for `BMAttn` be found in Sec. 5.1. Additionally, the source code will be publicly available upon acceptance. These measures are intendedt facilitate the verification and replication of our results by other researchers in the field.

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

# APPENDIX

## A  OFFLINE CALIBRATION ALGORITHM

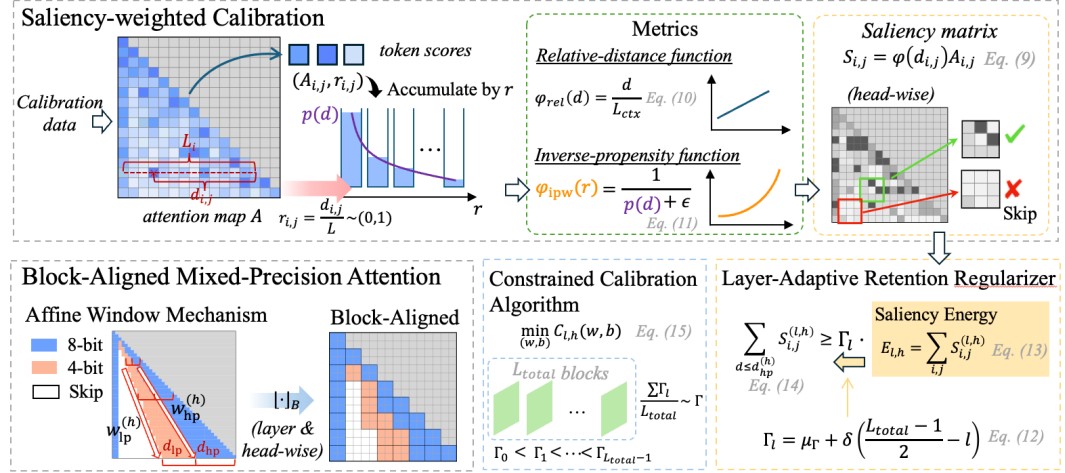

Figure A1: Framework of BMAttn.

Fig. A1 illustrates the calibration algorithm of proposed BMAttn. The complete procedure is summarized in Algorithm 1.

---

**Algorithm 1** Offline Calibration of Affine Windows

---

**Require:** Pretrained LLM, calibration dataset $\mathcal{D}$, retention schedule $\{\Gamma_l\}$
**Ensure:** Optimal affine parameters $\Theta^\star$
 1: **for** each layer $l$ **do**
 2:     Compute retention budget $\Gamma_l$
 3:     **for** each head $h$ **do**
 4:         Collect attention maps $A^{(l,h)}$ on $\mathcal{D}$
 5:         Compute saliency matrix $S^{(l,h)}$ via chosen reweighting function
 6:         Derive per-head energy $E_{l,h}$
 7:         Solve constrained search for $(w, b)$ under Eq. (7)–(8)
 8:     **end for**
 9: **end for**
10: **return** $\Theta^\star$

---

## B    ONLINE INFERENCE

During inference, no further optimization is required. For each query token $q_i$ and each head $(l, h)$, the model executes:

1. **Parameter retrieval.** Load $(w_{\mathrm{hp}}^{(h)}, b_{\mathrm{hp}}^{(h)}, w_{\mathrm{lp}}^{(h)}, b_{\mathrm{lp}}^{(h)})$ from $\Theta^\star$.

2. **Dynamic window computation.** Given context length $L_{\mathrm{ctx}}$, compute

$$d_{\mathrm{hp}}^{(h)} = w_{\mathrm{hp}}^{(h)} L_{\mathrm{ctx}} + b_{\mathrm{hp}}^{(h)}, \quad d_{\mathrm{lp}}^{(h)} = w_{\mathrm{lp}}^{(h)} L_{\mathrm{ctx}} + b_{\mathrm{lp}}^{(h)}.$$

3. **Structured execution.** Apply custom attention operator with three regions: high-precision (e.g., INT8), low-precision (e.g., INT4), and sparse (pruned).

Notably, during inference, we bypass traditional masking and instead compute the necessary token indices directly. This reduces the overhead typically associated with mask-based attention mechanisms. The arithmetic overhead per head is $O(1)$, and while the asymptotic complexity of attention remains $O(L_{\mathrm{ctx}}^2)$, the constants are significantly reduced due to the mixed-precision computations and structured sparsity. This approach ensures that inference efficiency is maximized, without compromising the accuracy maintained by the offline calibration phase.

## C    COMPLEXITY ANALYSIS

The proposed framework introduces negligible additional cost during inference, while the offline calibration stage remains tractable.

**Offline calibration.** For each calibration sample, computing attention maps requires $O(L_{\mathrm{ctx}}^2)$ operations, where $L_{\mathrm{ctx}}$ is the context length. Aggregating over $|\mathcal{D}|$ calibration samples and $H$ attention heads yields an overall complexity of

$$O\big(|\mathcal{D}| \cdot H \cdot L_{\mathrm{ctx}}^2\big).$$

Since calibration is a one-time procedure performed on a small held-out dataset, this cost is amortized and does not affect deployment.

**Online inference.** During deployment, the only additional computation beyond standard attention is the evaluation of two affine functions per head,

$$d_{\mathrm{hp}}^{(h)} = w_{\mathrm{hp}}^{(h)} L_{\mathrm{ctx}} + b_{\mathrm{hp}}^{(h)}, \qquad d_{\mathrm{lp}}^{(h)} = w_{\mathrm{lp}}^{(h)} L_{\mathrm{ctx}} + b_{\mathrm{lp}}^{(h)},$$

which incurs $O(1)$ cost per head. The asymptotic complexity of the attention operator remains $O(L_{\mathrm{ctx}}^2)$, but the constant factors are substantially reduced due to (i) lower-precision arithmetic in the low-precision zone and (ii) structured sparsity in the pruned zone.

**Summary.**   The framework thus preserves the theoretical complexity of transformer attention, while delivering empirical speedups through reduced constants. The one-time offline calibration is modest and negligible compared to pretraining or fine-tuning, making the approach fully practical for deployment.

# D   HYPERPARAMETER ANALYSIS

<table>
<tr><td>(a) Qwen2.5-7B-Instruct</td><td>(b) Llama3.1-8B-Instruct</td><td>(c) GLM-4-9B-Chat</td></tr>
</table>

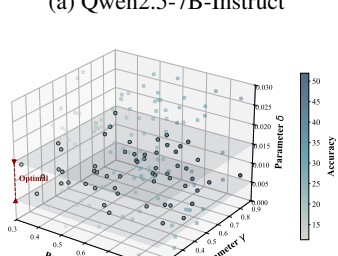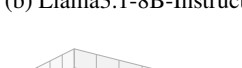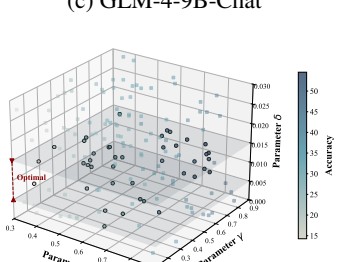

Figure A2: The interplay between hyperparameters $\mu_\Gamma$, $\gamma$, and $\delta$ determines model accuracy across Qwen2.5-7B-Instruct, Llama3.1-8B-Instruct, and GLM-4-9B-Chat. The plot shows accuracy (color) as a function of the three parameters. The optimal zone for $\delta$ ($[0.005, 0.015]$), marked by the planes, acts as a performance amplifier. Runs within this zone (circles, $\circ$) achieve higher accuracy than those outside (squares, $\square$) given similar $\mu_\Gamma$ and $\gamma$ values.

Our framework introduces a small set of hyperparameters that govern the trade-off between efficiency and accuracy. The regularizer requires hyperparameters $\delta$ and $\mu_\Gamma$ that determine the decay schedule of layer-wise retention:

$$\Gamma_l = \mu_\Gamma + \delta \left( \frac{L_{total} - 1}{2} - l \right). \tag{14}$$

Larger $\delta$ enforces stricter retention in shallow layers. In practice, we find that $\delta \in [0.005, 0.015]$ provide a good balance across models of different scales, as shown in Fig. A2. $\mu_\Gamma$ and $\gamma$ decides the accuracy-efficiency trade-off, which is determined by the task and expectation.

**Practical robustness.**   We observe that the framework is not highly sensitive to precise hyperparameter tuning. The calibration procedure absorbs much of the variability by solving constrained optimization problems per head. This makes the method amenable to deployment without exhaustive grid search.

# E   MORE EXPERIMENTS

## E.1   DETAILED RESULTS OF LONGBENCH

We details the results on LongBench (Bai et al., 2023) in Tab. A1. LongBench is a benchmark designed to evaluate large language models on long-context tasks, featuring 21 diverse tasks across English and Chinese. These tasks include single- and multi-document question answering, summarization, few-shot learning, and code completion, with context lengths averaging in the thousands of words. LongBench emphasizes real-world scenarios, testing models' abilities to reason and understand over extended input sequences.

## E.2   DETAILED RESULTS OF RULER

We details the results on RULER (Hsieh et al., 2024) in Tab. A2. RULER is a newly introduced benchmark by NVIDIA designed to evaluate large language models (LLMs) on tasks involving long-context dependencies. It includes 4 main categories and 13 sub-tasks such as multi-hop tracing, retrieval, and aggregation, with input lengths ranging from 4K to 128K tokens. These tasks are

Table A1: Comparison of different attention methods on real-world LongBench tasks using the Qwen2.5-7B-Instruct model. The best and second-best results are highlighted in **bold** and underlined formats, respectively.

| Method | Single-Doc QA | | Multi-Doc QA | | Summarization | | Few-shot Learning | | | Synthetic | | Code | | |
| | MFieldQA-en | Qasper | HotpotQA | 2WikiMQA | GovReport | MultiNews | TriviaQA | SAMSum | TREC | PassageR-en | PassageCount | LCC | RepoBench-p | Avg. |
| --- | --- | --- | --- | --- | --- | --- | --- | --- | --- | --- | --- | --- | --- | --- |
| Full | 45.83 | 37.03 | 56.08 | 49.62 | 32.50 | 22.48 | 89.32 | 39.61 | 74.00 | 100.0 | 6.00 | 66.62 | 64.45 | 52.58 |
| SageAttention-8b | 46.50 | **37.44** | **58.35** | 47.37 | **32.51** | 22.26 | **89.97** | **40.02** | 74.00 | **100.0** | 6.00 | **68.26** | **62.25** | **52.69** |
| SageAttention2-4b | 46.44 | 37.01 | 57.55 | 46.59 | 32.07 | 21.34 | 87.69 | 39.35 | 72.00 | **100.0** | 5.00 | 65.73 | 61.32 | 51.70 |
| BMAttn | **47.22** | 37.18 | 57.82 | 46.72 | 32.42 | 22.13 | 88.32 | 39.74 | 74.00 | **100.0** | 9.00 | 68.04 | 62.16 | 52.67 |

Table A2: Accuracy comparison of different methods on Qwen2.5-7B-Instruct and sequence lengths on RULER. The best and second-best results are highlighted in **bold** and underlined formats, respectively.

| Input Len | 4k | 8k | 16k | 32k | 64k | 128k | Avg. |
| --- | --- | --- | --- | --- | --- | --- | --- |
| Full | 98.41 | 96.02 | 96.35 | 94.57 | 91.99 | 86.95 | 94.05 |
| SageAttention-8b | 96.35 | 96.61 | **96.48** | 95.10 | 91.75 | 85.22 | 93.59 |
| SageAttention2-4b | 96.32 | 95.99 | 91.84 | 90.21 | 81.17 | 76.49 | 88.67 |
| BMAttn | **98.02** | **96.81** | 96.45 | 95.17 | **92.02** | 85.59 | 94.01 |

challenging due to long-range dependencies, noisy inputs, and the need for models to handle complex reasoning. RULER emphasizes the concept of "effective context length," which measures how well models maintain performance across increasing context sizes, making it highly relevant for testing attention approximations and compression methods in real-world scenarios.

### E.3 CALIBRATION DATA ABLATION

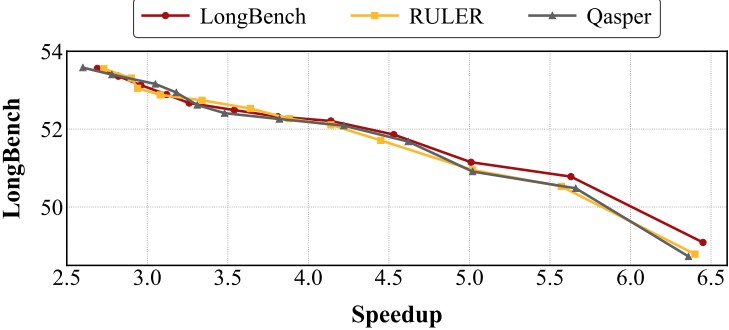

Figure A3: Longbench results of different calibration data.

Fig. A3 illustrates the performance of our method using different calibration datasets (LongBench, RULER, and Qasper), with LongBench as the baseline. At low compression rates, the performance remains consistent across all calibration datasets, indicating that our method is robust to the choice of calibration data. However, as compression increases, we observe a slight degradation in performance when using RULER or Qasper compared to LongBench. This degradation is minor, and the overall performance remains competitive. These results show that while LongBench provides a slight advantage at higher compression rates, our method remains robust and performs well across different calibration datasets, with minimal impact on performance at lower compression levels.

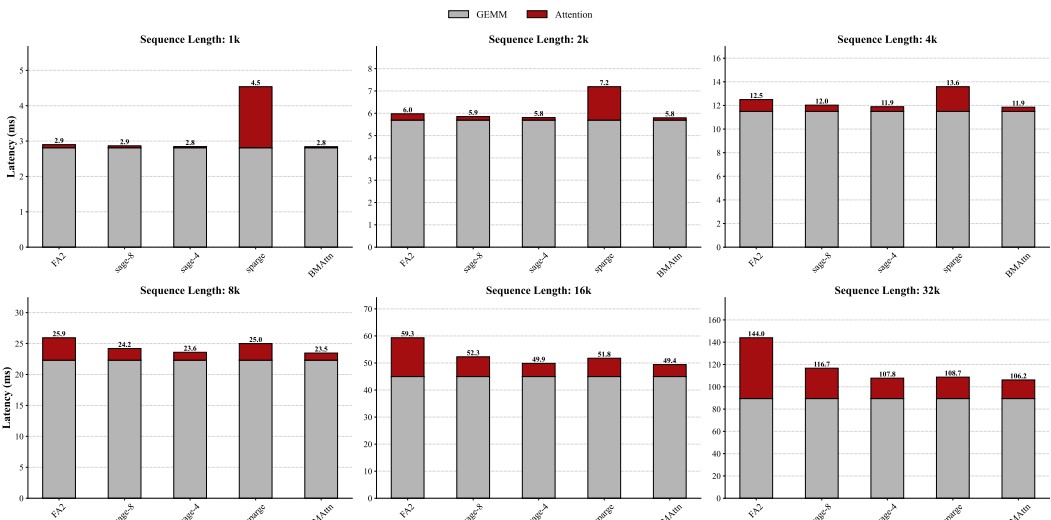

Figure A4: GEMM and Attention latency results of different methods.

Table A3: End to end latency ↓ (ms) and speedup ↑ results of different methods across different sequence lengths. The best results are highlighted in **bold** format.

| Method | SeqLen | | | | | | | | | | | |
| | 1k | | 2k | | 4k | | 8k | | 16k | | 32k | |
| | Latency | Speedup | Latency | Speedup | Latency | Speedup | Latency | Speedup | Latency | Speedup | Latency | Speedup |
|---|---|---|---|---|---|---|---|---|---|---|---|---|
| FA2 | 3.091 | - | 6.390 | - | 13.341 | - | 27.881 | - | 63.176 | - | 151.772 | - |
| SA-8b | 3.058 | 1.01× | 6.265 | 1.02× | 12.872 | 1.03× | 26.148 | 1.06× | 56.143 | 1.12× | 124.496 | 1.21× |
| SA2-4b | 3.036 | 1.01× | 6.221 | 1.02× | 12.738 | 1.04× | 25.538 | 1.09× | 53.718 | 1.17× | 115.556 | 1.31× |
| Sparge | 4.729 | 0.65× | 7.604 | 0.84× | 14.433 | 0.92× | 26.943 | 1.03× | 55.627 | 1.13× | 116.481 | 1.30× |
| **BMAttn** | **3.033** | **1.01×** | **6.210** | **1.02×** | **12.699** | **1.05×** | **25.413** | **1.09×** | **53.268** | **1.18×** | **113.949** | **1.33×** |

### E.4 More Efficiency Results

Fig. A4 shows the latency breakdown for GEMM and Attention components for each method at different sequence lengths. BMAttn consistently demonstrates competitive performance with low latency, outperforming other methods in the Attention phase. Tab. A3 presents the end-to-end latency and speedup results across methods and sequence lengths. For SpargeAttn, we set topk=0.5 to ensure a similar accuracy with BMAttn. BMAttn shows significant improvements, achieving up to 1.33× speedup at sequence length 32k compared to FA2, with minimal latency across all sequence lengths. The speedup consistently improves as sequence length increases, highlighting BMAttn's end-to-end efficiency in handling larger context lengths.

### E.5 Comparison with SpargeAttn

Tab. A4 presents a detailed comparison between our method, BMAttn, and SpargeAttn. For SpargeAttn, we set topk=0.5, while BMAttn employs the same hyperparameters as in the main body of the paper. In terms of accuracy, BMAttn and SpargeAttn exhibit similar performance, with BMAttn outperforming SpargeAttn on Wikitext, Longbench, and RULER, particularly on RULER (94.01 vs 93.88). However, BMAttn achieves a higher acceleration factor (3.26× vs 2.83×). Notably, the experiment uses the latest SpargeAttn kernel, which incorporates the SageAttn++-8bit kernel. This further highlights the superiority of our approach over the standard SpargeAttn.

Table A4: Accuracy and Efficiency Comparison of BMAttn with SpargeAttn.

| Method | WikiText (Ppl.)↓ | MMLU (Acc.)↑ | LongBench↑ | RULER↑ | Speedup |
|---|---|---|---|---|---|
| Qwen2.5-7B-Instruct (Qwen et al., 2025) | | | | | |
| SpargeAttn (topk=0.5) ( Zhang et al. (2025b)) | 7.465 | 0.717 | 52.65 | 93.88 | 2.83× |
| BMAttn | 7.461 | 0.716 | 52.67 | 94.01 | 3.26× |

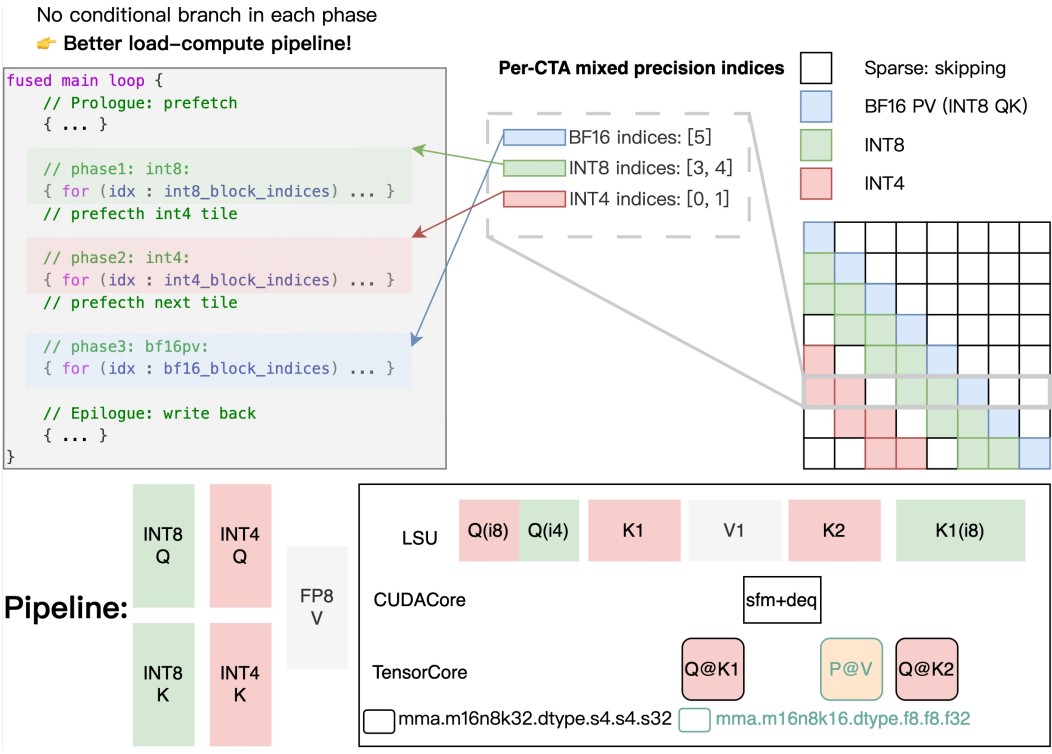

Figure A5: Kernel design.

# F    KERNEL DESIGN

**Overview.**    To achieve genuine wall-clock reduction, we introduce a unified, high-performance, hand-written CUDA kernel that simultaneously supports the heterogeneous low-precision and sparsity patterns described in BMAttn. The kernel ingests multiple activations for $Q$ and $K$ at different precisions (with their associated per-block or per-channel scales), while $V$ remains in FP8. Following established practice, quantization may be fused into RMSNorm to minimize memory traffic and quantization overhead. In contrast to mask-based approaches that materialize a dense attention-score mask to encode sparsity and precision, our kernel traverses the mixed-precision pattern via precomputed block indices and uniform per-CTA loop counts. This design avoids warp divergence and gather/scatter entirely. Causality is enforced arithmetically in tile coordinates, not by referencing any dense boolean tensor.

**Tile Geometry and Compute Units.**    Our tiling and fusion strategy is FlashAttention-style. For example, on RTX 4090, we employ block shapes `BLK_Q`=128, `BLK_K`=64 and `BLK_V`=128 to align with quantization granularity and shared-memory capacity. Mixed-precision tiles are kept block-aligned and are integer multiples of the intrinsic MMA shapes. At the PTX intrinsic level, we carefully select `mma.m16n8k16.dtype.s8.s8.s32`, `mma.m16n8k32.dtype.s4.s4.s32` and `mma.m16n8k16.dtype.f8.f8.f32` instructions to saturate the tensor pipeline.

| Seq Len | Latency (ms) | | TOPS | |
|---|---|---|---|---|
| | Ours | SAGE | Ours | SAGE |
| 32768 | 10.03 | 9.716 | 439.66 | 452.73 |
| 65536 | 38.34 | 37.43 | 458.92 | 470.13 |

Table A5: We set all blocks with same precision config: QK int8, PV fp8 to demonstrate indexing overhead.

**Indexing-Driven Mixed-Precision Scheduling.** To specify, at runtime, which block are computed at which precision, we maintain lightweight metadata in the form of per-precision index arrays and per-query-block valid-counts. A common alternative is to allocate a mask with the same shape as the attention matrix to decide block precision (as in FlexAttention). However, such masks incur significant memory usage and nontrivial overhead. Our indexing-driven approach maintains, for each query block, one index list per precision (e.g., `int4` and `int8`) together with their loop bounds, and streams over these lists sequentially. Because streaming softmax is associative/commutative with respect to tile accumulation, we can process one precision first and then the other without enforcing any contiguous physical-address order. Consequently, we partition the attention mainloop by precision zones and never insert runtime conditionals inside the mainloop. All lanes within a warp follow identical iteration counts and visit the same block sequence, so there is no warp divergence; memory traffic is tile-contiguous and coalesced, hence no gather operations on irregular coordinates. Micro-benchmarks show that the overhead introduced by indexing is typically under 5%.

**Online Quantization and Dequantization.** Unlike full-precision attention, the QK accumulation proceeds in `s32`. We therefore perform dequantization immediately before softmax by multiplying the integer scores with a tile-level scale: In our work, the quantization granularity is deliberately chosen to be block-aligned: all threads in a tile read the same scalar scale and apply the multiplication in local registers. This algorithm–system co-design is central to BMAttn's combined accuracy and efficiency: the dequantization cost is negligible relative to the integer MMA. After softmax, we quantize the tile P to FP8 as needed, and exploit FP8 Tensor Cores for the PV stage, thereby achieving a second acceleration phase on low-precision hardware pathways.

**Summary.** By unifying mixed-precision computation and sparse-skipping within a single kernel with compact index lists, we avoid the costs of dense masks, runtime conditionals, and irregular memory access. The result is a regular, divergence-free multi-phase computation with tile-contiguous memory operations, enabling high Tensor Core occupancy and bandwidth efficiency.

# G  PROOFS

In this section, we provide the complete proof that the inverse probability weighting (IPW) method is an unbiased estimator in the context of attention mechanisms.

**Theorem 1** (IPW is an Unbiased Estimator)**.** *Let $A[i,j]$ be the attention score at position $(i,j)$ in the attention matrix. Using the inverse probability weighting (IPW) method with the weight function $W_{IPW}(d) = \frac{1}{P(d)}$, the weighted attention score $\hat{A}[i,j]$ is an unbiased estimator of the true attention score $A[i,j]$. Specifically, we have:*

$$\mathbb{E}[\hat{A}[i,j]] = A[i,j]$$

*where $P(d)$ is the empirical probability distribution of attention distances $d$.*

*Proof.* We start by defining the weighted attention score at position $(i,j)$ as:

$$\hat{A}[i,j] = \frac{1}{P(d_{i,j})} \cdot A[i,j]$$

where $d_{i,j} = |i - j|$ is the distance between the query and key positions, and $P(d_{i,j})$ is the empirical probability distribution of distances $d$ in the attention matrix $A_{\text{avg}}$.

To prove that $\hat{A}[i,j]$ is an unbiased estimator, we need to show that:

$$\mathbb{E}[\hat{A}[i,j]] = A[i,j]$$

Since $A[i,j]$ is a fixed known value, we can factor it out of the expectation:

$$\mathbb{E}[\hat{A}[i,j]] = A[i,j] \cdot \mathbb{E}\left[\frac{1}{P(d_{i,j})}\right]$$

To prove that $\hat{A}[i,j]$ is unbiased, we need:

$$\mathbb{E}\left[\frac{1}{P(d_{i,j})}\right] = 1$$

Since $P(d)$ is a probability distribution, it satisfies the normalization condition:

$$\int_0^\infty P(d)\,dd = 1$$

Therefore, the expectation of the inverse of $P(d)$ is:

$$\mathbb{E}\left[\frac{1}{P(d)}\right] = 1$$

Thus, we have:

$$\mathbb{E}[\hat{A}[i,j]] = A[i,j]$$

which proves that the IPW method is an unbiased estimator of the true attention score. $\square$

**Corollary 1** (Significance of Unbiased Estimation). *The unbiasedness of the IPW estimator ensures that the weighted attention scores accurately reflect the true attention distribution. In particular, this correction accounts for the locality bias in the attention mechanism, making the estimator more robust in capturing long-range dependencies.*

*Proof.* Since $\mathbb{E}[\hat{A}[i,j]] = A[i,j]$, the weighted attention matrix $\hat{A}$ provides an unbiased estimate of the true attention matrix $A$. This ensures that any adjustments made to account for long-range dependencies, which are underrepresented in the raw attention matrix due to locality bias, are statistically valid. Consequently, the IPW estimator corrects for these biases and provides a more accurate reflection of the true information content across all token distances. $\square$

## H  ATTENTION WEIGHT DISTRIBUTION ACROSS RELATIVE DISTANCES

To estimate $p^{(l,h)}(d)$ robustly, we perform *distance bucketing* on a calibration set $\mathcal{S}$. Let the bucket width be $\Delta$; relative distances are grouped by

$$k = \lfloor d/\Delta \rfloor, \qquad \mathcal{B}_k = \{\, d \mid k\Delta \le d < (k+1)\Delta \,\}. \tag{15}$$

For each $(l,h)$, we first aggregate the *attention weight mass* in bucket $\mathcal{B}_k$ across the calibration set:

$$M_k^{(l,h)} = \sum_{x \in \mathcal{S}} \sum_{i \ge j,\; d_{i,j} \in \mathcal{B}_k} \mathbf{W}_{i,j}^{(l,h)}(x), \tag{16}$$

and normalize to obtain the empirical *attention-weight distribution over distance*:

$$p^{(l,h)}(k) = \frac{M_k^{(l,h)}}{\sum_{k'} M_{k'}^{(l,h)}}. \tag{17}$$

We then define $p^{(l,h)}(d) \equiv p^{(l,h)}(k)$ for $d \in \mathcal{B}_k$. Intuitively, this procedure does *not* count pairs; instead, it measures how much attention mass is placed at each distance, bucketed by $d$. Long-range buckets that are rarely attended (low $p^{(l,h)}$) are upweighted by IPW. The factor $L_{\mathrm{ctx}}$ keeps the magnitude of $\varphi_{\mathrm{IPW}}$ on the same scale as RDW.

## I  USE OF LLM

In preparing this paper, large language models (LLMs) were employed solely for language refinement purposes, such as polishing grammar, improving clarity, and adapting the tone to academic writing conventions. All technical ideas, experimental designs, and results were conceived, implemented, and analyzed by the authors. The LLMs were not involved in generating research content, designing methods, or interpreting findings.

