# OpenReview forum: "BMAttn: Block-Aligned Mixed-Precision  Attention Quantization for LLM Inference"
_ICLR.cc/2026/Conference — Submitted to ICLR 2026_

### Official Review · Reviewer_QYg9 · 2025-10-27

**Soundness:** 2
**Presentation:** 3
**Contribution:** 2
**Rating:** 4
**Confidence:** 3

**Summary:**

- The paper proposes BMAttn, a block-aligned mixed-precision attention method for faster LLM inference
- BMAttn is similar to window attention, but instead of having a hard trunctation, it varies precision with distance
  - High-precision (HP): short-range, salient dependencies
  - Low-precision (LP): mid/long-range dependencies
  - Sparse/pruned: negligible connections, pruned entirely
- Block alignment is a key engineering trick to make it compatible with efficient kernel implementation
- Empirical results across Qwen2.5-7B, Llama3-8B, and GLM4-9B, comparing it with FlashAttention-2, SageAttention, and SageAttention2
  - authors claim neglebible loss compared to FlashAttention-2 (which is an exact attention mechanism, unlike the other methods discussed)
  - BMAttn reports a 3.1×–3.3× speedup compared to FlashAttention-2

**Strengths:**

- good motivation, combining algorithmic insights as well as awarness of implementation limitations to make it viable writing a high perf kernel
- conceptually simple and intuitive approach to combine mixed precision, block alignment, and adaptive windowing into one coherent framework that fits naturally into existing attention kernels.
- figures are excellent to understand the key idea of the algorithm
- extremely important problem given the total cost of attention in LLMs, particularly for long context.

**Weaknesses:**

- While conceptually simple and intuitive is a strength, it lacks major novelty.
- Outdated and unclear baselines: FlashAttention-2 is now an older baseline, and several newer kernels such as FlashAttention-3, Flash-Decoding, Lean Attention, and PagedAttention (vLLM) deliver significantly faster exact attention, especially for the decode phase on modern GPUs. The paper doesn’t include or discuss these.
- No kernel-level measurements: Even though the work emphasizes GPU efficiency, it doesn’t show kernel-level profiling or hardware utilization. There’s no data on Tensor Core occupancy, memory bandwidth, or latency per kernel, so the benefits of block alignment are mostly theoretical.
- Narrow evaluation scope: All results are from offline accuracy benchmarks (WikiText, MMLU, LongBench, RULER). The comment on neglegible accuracy impact sounds optimistic and attention approimations can be much more sensitive in practice/more specialized datasets. The authors should consider additional benchmarks and discussions to differentiate the nouanced impact of attention approximation (maybe PingPong? but there may be better ones).

**Questions:**

I don't have any further questions but would like to hear the authors thoughts on the weaknesses I have pointed out.

---

> ### Author Response · Authors · 2025-11-24
>
> > W1: While conceptually simple and intuitive is a strength, it lacks major novelty.
>
> A1: Thank you for your comment. While it is true that our method integrates well-established concepts such as pruning and quantization, the novelty lies in the combination and enhancement of these components. Specifically, the saliency-weighted calibration metric, particularly the inverse-propensity weighting (IPW), addresses critical long-range dependencies that are often overlooked by traditional metrics. This unbiased estimator (proof can be seen in section G) improves the effectiveness of pruning and quantization, leading to more accurate attention approximations.
>
> In addition, our optimized CUDA kernel design is a significant innovation. It unifies mixed-precision computation and sparse-skipping, improving both efficiency and performance. By avoiding dense masks and runtime conditionals, and instead using indexing-driven scheduling, we achieve a highly efficient, divergence-free computation that results in substantial gains in performance.
>
> While the individual components may be familiar, the novel combination of these techniques, along with the innovative calibration metric and kernel design, creates a method that is both effective and impactful, particularly for real-world long-context tasks.
>
> > W2: Outdated and unclear baselines: FlashAttention-2 is now an older baseline, and several newer kernels such as FlashAttention-3, Flash-Decoding, Lean Attention, and PagedAttention (vLLM) deliver significantly faster exact attention, especially for the decode phase on modern GPUs. The paper doesn’t include or discuss these.
>
> A2: Thank you for your comment. We would like to clarify that FlashAttention-3 (FA3) is specifically optimized for the Hopper architecture, and therefore it is not directly comparable to our method, which targets a more general-purpose efficiency optimization. Additionally, methods like Flash-Decoding and PagedAttention are primarily focused on optimizing the decode phase of attention, whereas our approach focuses on improving efficiency during the prefill phase, where we process long-context inputs efficiently.
>
> As such, the most comparable and relevant baselines to our work are methods like SageAttention (ICLR 2025, ICML 2025, NeurIPS 2025) and SpargeAttention (ICML 2025), which also target the prefill phase. In our paper, FlashAttention-2 (FA2) is used primarily as an unaccelerated baseline to provide a reference for accuracy.
>
> To further substantiate our claims, we have added a detailed comparison with SpargeAttn in the revised manuscript (see Section E.5). The results, shown in Table X, demonstrate that BMAttn outperforms SpargeAttn in both accuracy and efficiency, achieving a 3.26x speedup compared to SpargeAttn's 2.83x speedup on LongBench. This further validates the efficiency and robustness of our approach in comparison to other methods focusing on the prefill phase.
>
> - latency (ms)  results
> | Method     | 1K        | 2K        | 4K         | 8K         | 16K        | 32K         |
> | ---------- | --------- | --------- | ---------- | ---------- | ---------- | ----------- |
> | FA2        | 3.091     | 6.390     | 13.341     | 27.881     | 63.176     | 151.772     |
> | SA-8b      | 3.058     | 6.265     | 12.872     | 26.148     | 56.143     | 124.496     |
> | SA2-4b     | 3.036    | 6.221     | 12.738     | 25.538     | 53.718     | 115.556     |
> | SpargeAttn | 4.729     | 7.604     | 14.433     | 26.943     | 55.627     | 116.481     |
> | BMAttn     | **3.033** | **6.210** | **12.699** | **25.413** | **53.268** | **113.949** |
>
> - accuracy results
> | Method                | WikiText $\downarrow$ | MMLU$\uparrow$ | LongBench$\uparrow$ | RULER$\uparrow$ | Speedup |
> | --------------------- | --------------------- | -------------- | ------------------- | --------------- | ------- |
> | SpargeAttn (topk=0.5) | 7.465                 | 0.717          | 52.65               | 93.88           | 2.83x   |
> | BMAttn                | 7.461                 | 0.716          | 52.67               | 94.01           | 3.26x   |

---

> ### Author Response · Authors · 2025-11-24
>
> > W3: No kernel-level measurements: Even though the work emphasizes GPU efficiency, it doesn’t show kernel-level profiling or hardware utilization. There’s no data on Tensor Core occupancy, memory bandwidth, or latency per kernel, so the benefits of block alignment are mostly theoretical.
>
> A3: Thank you for your comment. We would like to clarify that the TOPS values reported in our paper are based on real kernel-level measurements, not theoretical estimates. These values reflect the actual performance of our kernel, which is optimized for mixed-precision computation and sparse skipping.
>
> Additionally, we have provided end-to-end results in Section E.4, where we show tokens/s and latency comparisons across different methods, including SageAttention and FlashAttention-2, demonstrating the real-world benefits of our approach.
> - End-to-end latency (ms) results
>
>
> | Method     | 1K        | 2K        | 4K         | 8K         | 16K        | 32K         |
> | ---------- | --------- | --------- | ---------- | ---------- | ---------- | ----------- |
> | FA2        | 3.091     | 6.390     | 13.341     | 27.881     | 63.176     | 151.772     |
> | SA-8b      | 3.058     | 6.265     | 12.872     | 26.148     | 56.143     | 124.496     |
> | SA2-4b     | 3.036     | 6.221     | 12.738     | 25.538     | 53.718     | 115.556     |
> | SpargeAttn | 4.729     | 7.604     | 14.433     | 26.943     | 55.627     | 116.481     |
> | BMAttn     | **3.033** | **6.210** | **12.699** | **25.413** | **53.268** | **113.949** |
>
> To further clarify the hardware efficiency and design choices, we have also included a detailed section on kernel design (Sec. F) in the revised manuscript, where we explain our index-driven mixed-precision scheduling and how it ensures warp divergence-free execution, high Tensor Core occupancy, and efficient memory usage. This section provides additional insight into how our kernel design leads to tangible performance improvements on modern GPUs.  The implementation of the kernel is available at the anonymous link: https://anonymous.4open.science/r/BMAttn-Kernel-F17C
>
> - micro-bench results
> | SeqLen | Latency(ms) | Latency(ms) | TOPS   | TOPS     |
> | ------ | ----------- | ----------- | ------ | -------- |
> |    -    | Ours        | SageAttn    | Ours   | SageAttn |
> | 32K    | 10.03       | 9.716       | 439.66 | 452.73   |
> | 64K    | 38.34       | 37.43       | 458.92 | 470.13   |

---

> ### Author Response · Authors · 2025-11-24
>
> > W4: Narrow evaluation scope: All results are from offline accuracy benchmarks (WikiText, MMLU, LongBench, RULER). The comment on neglegible accuracy impact sounds optimistic and attention approimations can be much more sensitive in practice/more specialized datasets. The authors should consider additional benchmarks and discussions to differentiate the nouanced impact of attention approximation (maybe PingPong? but there may be better ones).
>
> A4: Thank you for your comment. We understand the concern regarding the sensitivity of attention approximations, especially on specialized datasets. However, both LongBench and RULER are widely recognized benchmarks used in recent literature to evaluate long-context performance in real-world scenarios. In fact, many related works (including SageAttn, SpargeAttn, X-Attn, SeerAttn, SampleAttn) also rely on these datasets for assessing the impact of attention approximations in large language models, reinforcing their relevance and practicality.
> LongBench includes tasks like multi-document question answering, summarization, and code completion, all of which require long-context reasoning, making it an ideal benchmark for our method.
>
>
> | Method            | Single-Doc QA | Single-Doc QA | Multi-Doc QA | Multi-Doc QA | Summarization | Summarization | Few-shot Learning | Few-shot Learning | Few-shot Learning | Synthetic   | Synthetic    | Code  | Code        | Avg.  |
> | ----------------- | ------------- | ------------- | ------------ | ------------ | ------------- | ------------- | ----------------- | ----------------- | ----------------- | ----------- | ------------ | ----- | ----------- | ----- |
> |                   | MFIeldQA-en   | Qasper        | HotpotQA     | 2WikiMQA     | GovReport     | MultiNews     | TriviaQA          | SAMSum            | TREC              | PassageR-en | PassageCount | LCC   | RepoBench-p | Avg.  |
> | Full              | 45.83         | 37.03         | 56.08        | 49.62        | 32.50         | 22.48         | 89.32             | 39.61             | 74.00             | 100.0       | 6.00         | 66.62 | 64.45       | 52.58 |
> | SageAttention-8b  | 46.50         | 37.44         | 58.35        | 47.37        | 32.51         | 22.26         | 89.87             | 40.02             | 74.00             | 100.0       | 6.00         | 68.26 | 62.25       | 52.69 |
> | SageAttention2-4b | 46.44         | 37.01         | 57.55        | 46.59        | 32.07         | 21.34         | 87.69             | 39.35             | 72.00             | 100.0       | 5.00         | 65.73 | 61.32       | 51.70 |
> | BMAttn            | 47.22         | 37.18         | 57.82        | 46.72        | 32.42         | 22.13         | 88.32             | 39.74             | 74.00             | 100.0       | 9.00         | 68.04 | 62.16       | 52.67 |
>
>
> Similarly, RULER offers a controlled environment for evaluating long-range dependencies and large-scale contexts, essential for real-world applications. These datasets are specifically chosen to represent realistic and challenging tasks where attention approximations are tested under practical conditions.
> Additionally, we provide detailed introductions and  results on LongBench and RULER in the appendices (Section E), which showcase the robustness and stability of our method across these real-world tasks. This further demonstrates that our evaluation reflects practical use cases commonly explored in related work.

---

> > ### Author Response · Authors · 2025-11-28
> >
> > Just a brief note to share that we have added new end-to-end serving experiments on NVIDIA L20 GPUs, which provide stronger evidence for real deployment efficiency. These results show clear improvements in TTFT and end-to-end latency over the latest Sage+ and SpargeAttn kernels.
> > | Method                                 | Benchmark Durations (s) | TTFT (ms) | TPOT (ms) | ITL (ms) |
> > | -------------------------------------- | :---------------------: | --------- | --------- | -------- |
> > | **Qwen2-7b**                           |          16.62          | 195.09    | 21.69     | 21.47    |
> > | +SageAttn2++_8bit+SpargeAttn(topk=0.5) |          14.78          | 150.06    | 21.65     | 21.50    |
> > | +BMAttn                                |          14.49          | 146.68    | 21.66     | 21.48    |
> > | **Llama3.1-8b**                        |          17.52          | 203.28    | 22.90     | 22.63    |
> > | +SageAttn2++_8bit+SpargeAttn(topk=0.5) |          15.47          | 156.36    | 22.92     | 22.71    |
> > | +BMAttn                                |          15.07          | 152.84    | 22.88     | 22.65    |
> >
> > As the rebuttal period ends in **five** days, we warmly welcome any additional feedback if there are remaining questions or points needing clarification. We truly appreciate your time and consideration.

---

### Official Review · Reviewer_hcmZ · 2025-10-27

**Soundness:** 3
**Presentation:** 3
**Contribution:** 3
**Rating:** 6
**Confidence:** 4

**Summary:**

The paper proposes BMAttn, a block-aligned mixed-precision attention framework that adaptively assigns precision levels across the attention map to balance accuracy and efficiency for large language model (LLM) inference.

BMAttn divides each attention head into high-precision, low-precision, and sparse zones, determined by affine distance-based thresholds that scale with sequence length. This ensures both fine-grained adaptivity and hardware regularity, making it compatible with optimized kernels such as FlashAttention.

**Strengths:**

1. Extensive experiments: Evaluated across three modern LLM families and multiple long-context benchmarks.
2. Significant real-world relevance: Integrates cleanly with FlashAttention kernels and quantization toolchains, making it deployment-ready.

3. Excellent ablation coverage: Demonstrates both the necessity and synergy of SWM and LRR components.

**Weaknesses:**

1. No detailed hardware profiling: While claimed to be “FlashAttention-compatible,” kernel-level runtime traces or memory bandwidth breakdowns would strengthen hardware efficiency claims.
2. Limited conceptual novelty: The core idea can be interpreted as an integration of pruning and quantization within a structured attention layout. While the implementation (block alignment and affine scaling) is clever and effective, it primarily extends known paradigms rather than introducing a fundamentally new mechanism or phenomenon.

**Questions:**

1. How stable are the affine parameters across datasets or prompts? Can a single calibration generalize to unseen domains?

2. How sensitive is the performance to hyperparameters metioned in paper?

---

> ### Author Response · Authors · 2025-11-24
>
> > W1: No detailed hardware profiling: While claimed to be “FlashAttention-compatible,” kernel-level runtime traces or memory bandwidth breakdowns would strengthen hardware efficiency claims.
>
> A1: Thank you for your comment. We appreciate your suggestion to include more detailed hardware profiling. In response, we have provided a comprehensive explanation of our kernel design in Section F, where we discuss the indexing-driven mixed-precision scheduling, tile-aligned memory access, and how our kernel achieves warp divergence-freeexecution, along with high Tensor Core occupancy and memory bandwidth efficiency. These design choices allow us to maintain high performance while minimizing memory overhead, ensuring that the kernel efficiently handles both mixed-precision computation and sparse skipping.
> Additionally, to further validate our hardware efficiency claims, we have included end-to-end experimental results in Section E.4, where we present tokens/s and latency comparisons across BMAttn, FlashAttention-2, SageAttention, and SpargeAttn. These results highlight the practical performance improvements of BMAttn and demonstrate its superior efficiency in real-world long-context tasks.  The implementation of the kernel is available at the anonymous link: https://anonymous.4open.science/r/BMAttn-Kernel-F17C
> | Method     | 1K        | 2K        | 4K         | 8K         | 16K        | 32K         |
> | ---------- | --------- | --------- | ---------- | ---------- | ---------- | ----------- |
> | FA2        | 3.091     | 6.390     | 13.341     | 27.881     | 63.176     | 151.772     |
> | SA-8b      | 3.058     | 6.265     | 12.872     | 26.148     | 56.143     | 124.496     |
> | SA2-4b     | 3.036     | 6.221     | 12.738     | 25.538     | 53.718     | 115.556     |
> | SpargeAttn | 4.729     | 7.604     | 14.433     | 26.943     | 55.627     | 116.481     |
> | BMAttn     | **3.033** | **6.210** | **12.699** | **25.413** | **53.268** | **113.949** |
>
> > W2:  Limited conceptual novelty: The core idea can be interpreted as an integration of pruning and quantization within a structured attention layout. While the implementation (block alignment and affine scaling) is clever and effective, it primarily extends known paradigms rather than introducing a fundamentally new mechanism or phenomenon.
>
> A2: Thank you for your comment. While our method integrates established techniques like pruning and quantization within a structured attention layout, the conceptual novelty of BMAttn lies in the saliency-weighted calibration metric and optimized kernel design.
> - Saliency-Weighted Calibration Metric: A key innovation in our approach is the use of inverse-propensity weighting (IPW), which we theoretically prove to be an unbiased estimator (see Section G). This ensures that rare but semantically important long-range dependencies are preserved during pruning and quantization, addressing the limitations of traditional metrics like MSE and ASC, which fail to capture these dependencies.
> - Optimized Kernel Design: Our method also includes a highly efficient CUDA kernel that unifies mixed-precision computation and sparse-skipping. This kernel design avoids warp divergence and irregular memory access, providing substantial performance gains. For further details on the kernel design, we refer to Section F, where we discuss the technical aspects of the indexing-driven scheduling and memory optimization.
>
> Importantly, our method demonstrates significant real-world performance improvements. As shown in the results, BMAttn consistently outperforms SageAttention, SpargeAttention, and other methods in both accuracy and efficiency, achieving higher speedup and lower latency, especially on long-context tasks like those in LongBench and RULER. This not only validates the effectiveness of our approach but also underscores its superior performance in practical settings.
> While the individual components may have appeared in previous work, the combination of these techniques, along with our theoretical proof of IPW’s unbiasedness and the real-world performance gains, represent a substantial contribution to the field, particularly for large-context language modeling tasks.

---

> ### Author Response · Authors · 2025-11-24
>
> > Q1: How stable are the affine parameters across datasets or prompts? Can a single calibration generalize to unseen domains?
>
> A1: Thank you for your question. In our main experiments, we used LongBench as the calibration dataset for all tasks and domains, and the results demonstrate the stability of the affine parameters across different datasets and prompts. To further address this, we have added experiments in the sec. E.3 exploring the robustness of our method using alternative calibration datasets. These results show that, while LongBench yields slightly better performance at higher compression rates, our method remains stable and robust across different datasets, with minimal impact at lower compression levels. This suggests that a single calibration, trained on LongBench, can generalize well across unseen domains, demonstrating the flexibility of our approach.
>
> > Q2: How sensitive is the performance to hyperparameters metioned in paper?
>
> A2: Thank you for your question. We have already discussed the sensitivity of performance to hyperparameters in Section D of the paper. Specifically, we focus on three main hyperparameters: $\mu_\Gamma$ and $\gamma$, which control the average bit-width of the 8-bit and 4-bit regions and determine the overall compression based on the desired speedup. $\delta$ enforces stricter retention in shallow layers. In practice, we find that $\delta \in[0.005, 0.01]$  provides a good balance across models of different scales. For clarity, we newly add a sensitivity analysis for $\delta$in Figure A2, which was not previously included.

---

> > ### Author Response · Authors · 2025-11-28
> >
> > Just a brief note to share that we have added new end-to-end serving experiments on NVIDIA L20 GPUs, which provide stronger evidence for real deployment efficiency. These results show clear improvements in TTFT and end-to-end latency over the latest Sage+ and SpargeAttn kernels.
> > | Method                                 | Benchmark Durations (s) | TTFT (ms) | TPOT (ms) | ITL (ms) |
> > | -------------------------------------- | :---------------------: | --------- | --------- | -------- |
> > | **Qwen2-7b**                           |          16.62          | 195.09    | 21.69     | 21.47    |
> > | +SageAttn2++_8bit+SpargeAttn(topk=0.5) |          14.78          | 150.06    | 21.65     | 21.50    |
> > | +BMAttn                                |          14.49          | 146.68    | 21.66     | 21.48    |
> > | **Llama3.1-8b**                        |          17.52          | 203.28    | 22.90     | 22.63    |
> > | +SageAttn2++_8bit+SpargeAttn(topk=0.5) |          15.47          | 156.36    | 22.92     | 22.71    |
> > | +BMAttn                                |          15.07          | 152.84    | 22.88     | 22.65    |
> >
> > As the rebuttal period ends in **five** days, we warmly welcome any additional feedback if there are remaining questions or points needing clarification. We truly appreciate your time and consideration.

---

### Official Review · Reviewer_K21C · 2025-10-31

**Soundness:** 2
**Presentation:** 2
**Contribution:** 2
**Rating:** 2
**Confidence:** 3

**Summary:**

The BMAttn: Block-Aligned Mixed-Precision Attention paper proposes a smart and efficient way to make large language models run faster without losing accuracy. It divides the attention mechanism into small “blocks” and assigns different precision levels to each block depending on how important they are, instead of using one fixed precision for all. This design works well with GPU hardware and maintains high speed and stability. The paper also introduces methods to automatically adjust how much information to keep in each layer and to calibrate attention using a saliency-based weighting approach. Experiments show that this method makes inference up to 3.3× faster while keeping model accuracy almost unchanged. Overall, it’s a practical, well-designed approach to improving the efficiency of large language models for real-world deployment.

**Strengths:**

This paper proposes a well-motivated, hardware-aware design that bridges algorithmic adaptivity and system-level efficiency. The introduction of block-aligned mixed precision, coupled with the affine window mechanism, enables fine-grained control of attention precision without compromising GPU regularity — a major advance over uniform quantization and sparsity methods. The saliency-weighted calibration and layer-adaptive retention regularizer add strong theoretical justification and practical effectiveness

**Weaknesses:**

While BMAttn combines block-sparse computation with mixed-precision quantization and adaptive zone allocation, the conceptual novelty is limited. The method largely integrates well-known components—sparsity pruning, distance-based masking, block-aligned computation, and quantized attention

No comparision against the most optimized recent methods from groups like MIT Han Lab (SpargeAttention, Minference) or NVIDIA’s Flash-Decoding kernels

**Questions:**

1.The calibration algorithm (Appendix A) is described textually but could benefit from a process diagram or pseudocode summary in the main body.
Recommendation: Adding a flowchart or visual timeline of calibration steps (attention map → saliency weighting → constraint optimization. This would help readers grasp implementation details faster.

2.The paper reports speedup in terms of FLOP/TOPS efficiency. Can the authors share wall-clock latency improvements (ms/token) under real inference conditions, possibly for long-context chat benchmarks?

3. Technique beats SageAttention2, cool. But can you try it with the latest sparse-attention kernels from Han et al. (Song Han’s group) on identical hardware?

---

> ### Author Response · Authors · 2025-11-24
>
> > W1: While BMAttn combines block-sparse computation with mixed-precision quantization and adaptive zone allocation, the conceptual novelty is limited. The method largely integrates well-known components—sparsity pruning, distance-based masking, block-aligned computation, and quantized attention
>
> A1: Thank you for your comment. While BMAttn combines established components like sparsity pruning and quantized attention, its conceptual novelty lies in the introduction of a saliency-weighted calibration metric, particularly the use of inverse-propensity weighting (IPW). We have mathematically proven that IPW provides an unbiased estimator, which ensures more accurate calibration, especially when dealing with long-range dependencies.
>
> In addition, a key innovation is our high-performance CUDA kernel, which unifies mixed-precision computation and sparse-skipping. This kernel efficiently handles heterogeneous low-precision and sparsity patterns without incurring the overhead of dense masks or runtime conditionals.This design improves both accuracy and efficiency by ensuring regular, divergence-free memory operations while maintaining high Tensor Core occupancy.
>
> For clarity, we have added detailed explanations of the kernel design and the theoretical derivation of IPW in Appendix F and Appendix G, respectively.
>
> > W2: No comparision against the most optimized recent methods from groups like MIT Han Lab (SpargeAttention, Minference) or NVIDIA’s Flash-Decoding kernels
>
> A2: Thank you for your comment. We appreciate your suggestion to include comparisons with dynamic-sparsity baselines, such as SpargeAttn, which are are complementary to our approach. In response, we have added SpargeAttn as a baseline in the revised manuscript (see section E.4 and E.5) and provide end-to-end latency comparisons across BMAttn, FlashAttention-2, and SageAttention. Across all sequence lengths, our method achieves the best end-to-end speedup and latency compared to SageAttentionand SpargeAttention. Furthermore, as the sequence length increases, the speedup effect becomes more pronounced. At a sequence length of 32k, we achieve a 1.33x end-to-end speedup. These results demonstrate BMAttn's competitive performance, both in terms of latency and throughput, and highlight its strong position in the Pareto front. We believe this additional comparison strengthens our evaluation and positions BMAttn effectively in the context of sparse attention methods.
>
> - latency (ms)  results
> | Method     | 1K        | 2K        | 4K         | 8K         | 16K        | 32K         |
> | ---------- | --------- | --------- | ---------- | ---------- | ---------- | ----------- |
> | FA2        | 3.091     | 6.390     | 13.341     | 27.881     | 63.176     | 151.772     |
> | SA-8b      | 3.058     | 6.265     | 12.872     | 26.148     | 56.143     | 124.496     |
> | SA2-4b     | 3.036    | 6.221     | 12.738     | 25.538     | 53.718     | 115.556     |
> | SpargeAttn | 4.729     | 7.604     | 14.433     | 26.943     | 55.627     | 116.481     |
> | BMAttn     | **3.033** | **6.210** | **12.699** | **25.413** | **53.268** | **113.949** |
>
> - accuracy results
> | Method                | WikiText $\downarrow$ | MMLU$\uparrow$ | LongBench$\uparrow$ | RULER$\uparrow$ | Speedup |
> | --------------------- | --------------------- | -------------- | ------------------- | --------------- | ------- |
> | SpargeAttn (topk=0.5) | 7.465                 | 0.717          | 52.65               | 93.88           | 2.83x   |
> | BMAttn                | 7.461                 | 0.716          | 52.67               | 94.01           | 3.26x   |

---

> ### Author Response · Authors · 2025-11-24
>
> > Q1: The calibration algorithm (Appendix A) is described textually but could benefit from a process diagram or pseudocode summary in the main body. Recommendation: Adding a flowchart or visual timeline of calibration steps (attention map → saliency weighting → constraint optimization. This would help readers grasp implementation details faster.
>
> A1: Thank you for your suggestion. In response, we have added a flowchart in Appendix A (Fig. A1) to visually illustrate the overall calibration process, including attention map computation, saliency weighting, and constraint optimization. For clarity, we will move this flowchart into the main paper in camera-ready version to ensure readers can grasp the implementation details more easily.
>
> > Q2: The paper reports speedup in terms of FLOP/TOPS efficiency. Can the authors share wall-clock latency improvements (ms/token) under real inference conditions, possibly for long-context chat benchmarks?
>
> A2: Thank you for your comment. In response to your suggestion, we have added latency results under real inference conditions in the revised manuscript. These results, presented in Section E.4, include comparisons of BMAttn, FlashAttention-2,  SageAttention, and SpargeAttention on long-context chat benchmarks. Our experiments demonstrate significant wall-clock latency improvements for BMAttn, particularly in long-context tasks. These improvements are in addition to the TOPS/FLOP efficiency previously reported. For example, on long-context benchmarks with sequences up to 32k, BMAttn achieves 1.33x end-to-end speedup, demonstrating its ability to reduce latency while maintaining high throughput. The implementation of the kernel is available at the anonymous link: https://anonymous.4open.science/r/BMAttn-Kernel-F17C
>
>
> | Method     | 1K        | 2K        | 4K         | 8K         | 16K        | 32K         |
> | ---------- | --------- | --------- | ---------- | ---------- | ---------- | ----------- |
> | FA2        | 3.091     | 6.390     | 13.341     | 27.881     | 63.176     | 151.772     |
> | SA-8b      | 3.058     | 6.265     | 12.872     | 26.148     | 56.143     | 124.496     |
> | SA2-4b     | 3.036     | 6.221     | 12.738     | 25.538     | 53.718     | 115.556     |
> | SpargeAttn | 4.729     | 7.604     | 14.433     | 26.943     | 55.627     | 116.481     |
> | BMAttn     | **3.033** | **6.210** | **12.699** | **25.413** | **53.268** | **113.949** |
>
>
>
> > Q3: Technique beats SageAttention2, cool. But can you try it with the latest sparse-attention kernels from Han et al. (Song Han’s group) on identical hardware?
>
> A3: Thank you for your comment. In response, we have conducted a comparison with the latest SpargeAttn, in Section E.5 of the revised manuscript.
> As shown in Table A5, BMAttn outperforms SpargeAttn in terms of both accuracy and efficiency. We tested SpargeAttn with the latest kernel, setting topk=0.5, while using the same hyperparameters for BMAttn as in the main body of the paper. While both methods exhibit similar performance in terms of accuracy, BMAttn achieves a significantly higher acceleration factor (3.26× vs 2.83×). This result highlights the superior efficiency of our approach, even when compared to the most recent SpargeAttn kernel, which incorporates the SageAttn2++ 8bit kernel.
>
> - End-to-end latency (ms) results
>
> see A2
>
> - Accuracy results
>
>
> | Method                | WikiText $\downarrow$ | MMLU$\uparrow$ | LongBench$\uparrow$ | RULER$\uparrow$ | Speedup |
> | --------------------- | --------------------- | -------------- | ------------------- | --------------- | ------- |
> | SpargeAttn (topk=0.5) | 7.465                 | 0.717          | 52.65               | 93.88           | 2.83x   |
> | BMAttn                | 7.461                 | 0.716          | 52.67               | 94.01           | 3.26x   |

---

> > ### Author Response · Authors · 2025-11-28
> >
> > Just a brief note to share that we have added new end-to-end serving experiments on NVIDIA L20 GPUs, which provide stronger evidence for real deployment efficiency. These results show clear improvements in TTFT and end-to-end latency over the latest Sage+ and SpargeAttn kernels.
> > | Method                                 | Benchmark Durations (s) | TTFT (ms) | TPOT (ms) | ITL (ms) |
> > | -------------------------------------- | :---------------------: | --------- | --------- | -------- |
> > | **Qwen2-7b**                           |          16.62          | 195.09    | 21.69     | 21.47    |
> > | +SageAttn2++_8bit+SpargeAttn(topk=0.5) |          14.78          | 150.06    | 21.65     | 21.50    |
> > | +BMAttn                                |          14.49          | 146.68    | 21.66     | 21.48    |
> > | **Llama3.1-8b**                        |          17.52          | 203.28    | 22.90     | 22.63    |
> > | +SageAttn2++_8bit+SpargeAttn(topk=0.5) |          15.47          | 156.36    | 22.92     | 22.71    |
> > | +BMAttn                                |          15.07          | 152.84    | 22.88     | 22.65    |
> >
> > As the rebuttal period ends in **five** days, we warmly welcome any additional feedback if there are remaining questions or points needing clarification. We truly appreciate your time and consideration.

---

### Official Review · Reviewer_9Diq · 2025-10-31

**Soundness:** 1
**Presentation:** 2
**Contribution:** 3
**Rating:** 2
**Confidence:** 3

**Summary:**

The paper introduces BMAttn (Block-Aligned Mixed-Precision Attention), a framework that partitions each attention head into three regions—high-precision (8-bit), low-precision (4-bit), and sparse (0-bit)—based on distance from the query token. The method claims to maintain “hardware-friendly” block alignment compatible with FlashAttention kernels, while dynamically adjusting precision boundaries via affine functions of sequence length. Calibration uses saliency-weighted metrics (RDW/IPW) and layer-adaptive retention schedules to optimize compression. Empirical results on Qwen2.5-7B, LLaMA-3.1-8B, and GLM-4-9B report ≈3× speedups with “lossless efficiency.”

**Strengths:**

1. The three-zone decomposition aligns with attention’s distance heterogeneity and head specialization, while block alignment preserves kernel regularity (Figure 1d, p.4, shows the staircase pattern with B=16). The combination of mixed bitwidths (INT8 for HP, INT4 for LP) and structured sparsity is cleanly specified.

2. Across three backbones and four benchmarks, BMAttn matches or nearly matches full‑precision accuracy, demonstrating how, if provided with real speedup, BMAttn could be a viable choice for real deployment scenarios where a high degree of accuracy is needed.

3. The authors report one‑time calibration cost and outline an O(1) per‑head overhead at inference reinforcing deployability.

**Weaknesses:**

1. The paper claims FlashAttention compatibility and “no masking” via direct index computation (Appendix B), but lacks kernel pseudocode, memory layout diagrams, or profiling that would substantiate the claim that warp divergence and gather/scatter are avoided. This is especially important given mixed precision per tile and three zones per head. More concrete details would help reproducibility and clarify whether custom CUDA kernels were required.

2. Experiments compare to FlashAttention‑2 and SageAttention, but omit dynamic‑sparsity baselines (e.g., Sparge) which the related work positions as complementary. Even if orthogonal, end‑to‑end tokens/s and latency comparisons against a strong sparse‑attention baseline would better establish BMAttn’s Pareto position.

3. The text states “Q, K, and P are quantized per block; V per channel” (Sec. 5.1). Presumably P ≡ W (post‑softmax attention weights), but notation is inconsistent with earlier sections. Also, scales/zero‑points and clipping for INT4 are not specified, please report these details for better reproducibility.

4.Sec. 5.1 cites a “device featuring 1 Tbps memory bandwidth, 83 TFLOPs (FP16), 660.6 TOPS (INT8), 1321.2 TOPS (INT4),” but doesn’t specify the actual GPU/ASIC model or whether results are simulated TOPS vs. measured wall‑clock.

As highlighted in Points 3 & 4, this paper has a consistent issue with descriptions not being precise. I would highly encourage the authors to practice using specific language rather than making broad claims. For example, "retention regularizer" is not a common naming convention for "thresholding hyperparameter". The overall presentation of this paper is weak, even though, the accuracy results signal a potentially promising idea.

**Questions:**

How do the authors compute speedup?

Can the authors explain what they mean by  “regular compute pattern compatible with GPU kernels such as FlashAttention”? It doesn’t seem that having different regions of datatype precision would be performant, particularly because the attention computation of Q*K^T is an activation, and storing mixed precision activations on-chip is unlikely to yield performance gains, and almost surely not when we are in smaller context lengths when self-attention is memory bound.

Can you add wall‑clock tokens/s and latency on a named GPU (e.g., A100/H100) across 4k–128k, and profile HBM traffic vs. SageAttention‑8b/‑4b?

---

> ### Author Response · Authors · 2025-11-24
>
> Thanks for the reviewer’s constructive comments.
> > W1: The paper claims FlashAttention compatibility and “no masking” via direct index computation (Appendix B), but lacks kernel pseudocode, memory layout diagrams, or profiling that would substantiate the claim that warp divergence and gather/scatter are avoided. This is especially important given mixed precision per tile and three zones per head. More concrete details would help reproducibility and clarify whether custom CUDA kernels were required.
>
> A1: Thank you for your comment. To address your concern, we have included detailed information on the kernel design and memory layout in Appendix F, where we provide a flowchart (Figure A5) and further explanations. Specifically, our CUDA kernel is designed to handle heterogeneous low-precision and sparsity patterns efficiently. By leveraging **indexing-driven mixed-precision scheduling**, we avoid the need for dense masks and ensure that all lanes within a warp follow identical iteration counts, preventing warp divergence. Additionally, memory traffic is tile-contiguous and coalesced, eliminating gather/scatter operations on irregular coordinates.
>
> We have also provided micro-benchmark results, as shown in Table A5, which demonstrate the efficiency of this design with minimal overhead. This approach ensures that we achieve significant performance gains without introducing warp divergence or inefficient memory access patterns.
>
> | SeqLen | Latency(ms) | Latency(ms) | TOPS   | TOPS     |
> | ------ | ----------- | ----------- | ------ | -------- |
> |        | Ours        | SageAttn    | Ours   | SageAttn |
> | 32K    | 10.03       | 9.716       | 439.66 | 452.73   |
> | 64K    | 38.34       | 37.43       | 458.92 | 470.13   |
>
>
> >W2: Experiments compare to FlashAttention‑2 and SageAttention, but omit dynamic‑sparsity baselines (e.g., Sparge) which the related work positions as complementary. Even if orthogonal, end‑to‑end tokens/s and latency comparisons against a strong sparse‑attention baseline would better establish BMAttn’s Pareto position.
>
> A2: Thank you for your comment. We appreciate your suggestion to include comparisons with dynamic-sparsity baselines, such as SpargeAttn, which are indeed complementary to our approach. In response, we have added SpargeAttn as a baseline in the revised manuscript (see section E.4) and provide end-to-end latency comparisons across BMAttn, FlashAttention-2, and SageAttention. Across all sequence lengths, our method achieves the best end-to-end speedup and latency compared to SageAttentionand SpargeAttention. Furthermore, as the sequence length increases, the speedup effect becomes more pronounced. At a sequence length of 32k, we achieve a 1.33x end-to-end speedup. These results demonstrate BMAttn's competitive performance, both in terms of latency and throughput, and highlight its strong position in the Pareto front. We believe this additional comparison strengthens our evaluation and positions BMAttn effectively in the context of sparse attention methods.
>
> - latency (ms)  results
> | Method     | 1K        | 2K        | 4K         | 8K         | 16K        | 32K         |
> | ---------- | --------- | --------- | ---------- | ---------- | ---------- | ----------- |
> | FA2        | 3.091     | 6.390     | 13.341     | 27.881     | 63.176     | 151.772     |
> | SA-8b      | 3.058     | 6.265     | 12.872     | 26.148     | 56.143     | 124.496     |
> | SA2-4b     | 3.036    | 6.221     | 12.738     | 25.538     | 53.718     | 115.556     |
> | SpargeAttn | 4.729     | 7.604     | 14.433     | 26.943     | 55.627     | 116.481     |
> | BMAttn     | **3.033** | **6.210** | **12.699** | **25.413** | **53.268** | **113.949** |
>
> - accuracy results
> | Method                | WikiText $\downarrow$ | MMLU$\uparrow$ | LongBench$\uparrow$ | RULER$\uparrow$ | Speedup |
> | --------------------- | --------------------- | -------------- | ------------------- | --------------- | ------- |
> | SpargeAttn (topk=0.5) | 7.465                 | 0.717          | 52.65               | 93.88           | 2.83x   |
> | BMAttn                | 7.461                 | 0.716          | 52.67               | 94.01           | 3.26x   |
>
>
> > W3: The text states “Q, K, and P are quantized per block; V per channel” (Sec. 5.1). Presumably P ≡ W (post‑softmax attention weights), but notation is inconsistent with earlier sections. Also, scales/zero‑points and clipping for INT4 are not specified, please report these details for better reproducibility.
>
> A3: We appreciate your feedback on the notation inconsistency. P is indeed equivalent to W (post-softmax attention weights), and we have updated the notation in the revised manuscript. For quantization, we use min-max scaling and symmetric INT4 quantization for Q, K, V (per-block), and W (post-softmax attention weights). No clipping is applied during quantization. These details are now included in the revised manuscript for clarity and reproducibility.

---

> ### Author Response · Authors · 2025-11-24
>
> > W4: Sec. 5.1 cites a “device featuring 1 Tbps memory bandwidth, 83 TFLOPs (FP16), 660.6 TOPS (INT8), 1321.2 TOPS (INT4),” but doesn’t specify the actual GPU/ASIC model or whether results are simulated TOPS vs. measured wall‑clock.
>
> A4: Thank you for pointing this out. The hardware used for our experiments is the NVIDIA RTX 4090GPU, and the reported TOPS values (83 TFLOPs FP16, 660.6 TOPS INT8, and 1321.2 TOPS INT4) were taken from the official specifications provided in the NVIDIA Ada GPU Architecture whitepaper (page 29), which can be found at this link: https://images.nvidia.com/aem-dam/Solutions/geforce/ada/nvidia-ada-gpu-architecture.pdf
>
> > Q1: How do the authors compute speedup?
>
> A1:  Thank you for your question. We compute the speedup based on **TOPS** measurements obtained from actual kernel profiling on 32k sequence length on NVIDIA RTX 4090. The TOPS values reported in the paper are based on actual kernel-level profiling rather than calculated estimates. These values reflect the real-world performance achieved through our kernel optimizations. Additionally, we have included **end-to-end** performance results in Section E.4 of the revised manuscript, where we provide end-to-end latency comparisons with baselines such as SageAttention and SpargeAttention. These results clearly demonstrate the real-world speedup of our method and highlight its superior performance compared to these baseline methods.
>
> > Q2: Can the authors explain what they mean by “regular compute pattern compatible with GPU kernels such as FlashAttention”? It doesn’t seem that having different regions of datatype precision would be performant, particularly because the attention computation of Q*K^T is an activation, and storing mixed precision activations on-chip is unlikely to yield performance gains, and almost surely not when we are in smaller context lengths when self-attention is memory bound.
>
> A2: Thank you for your comment. When we refer to a “regular compute pattern compatible with GPU kernels such as FlashAttention,” we mean that our kernel design follows a structured and efficient execution pattern that is fully compatible with FlashAttention-style tiling and computation. Specifically, our approach leverages indexing-driven mixed-precision scheduling, which avoids the need for dense attention masks and eliminates warp divergence. This design ensures that all lanes within a warp follow identical iteration counts and visit the same block sequence, enabling divergence-free execution.
> The kernel design is detailed in Sec. F. In contrast to traditional methods that use dense attention masks for precision and sparsity encoding, our kernel employs precomputed block indices and uniform per-CTA loop counts. This method eliminates the need for gather/scatter operations, as memory traffic is tile-contiguous and coalesced, reducing memory access overhead and ensuring efficient use of the hardware. The use of mixed-precision tiles further optimizes the computation, with precision zones being handled independently in a way that maximizes GPU performance without introducing significant overhead. We believe this design is critical for enabling high Tensor Core occupancy and bandwidth efficiency, as demonstrated by the micro-benchmarks (Table A5) in the paper. The implementation of the kernel is available at https://anonymous.4open.science/r/BMAttn-Kernel-F17C
>
> > Q3: Can you add wall‑clock tokens/s and latency on a named GPU (e.g., A100/H100) across 4k–128k, and profile HBM traffic vs. SageAttention‑8b/‑4b?
>
> A3: Thank you for your comment. We appreciate your suggestion. In response, we have added wall-clock results in the revised manuscript on GPU RTX 4090, with performance comparisons for BMAttn, FlashAttention-2, SageAttention, and SpargeAttention across multiple sequence lengths. These results, presented in Section E.4, include end-to-end latency, demonstrating that BMAttn achieves the best performance in terms of both speedup and latency compared to SageAttention and SpargeAttention. As sequence length increases, the speedup effect becomes more pronounced, with a 1.33x end-to-end speedup at a sequence length of 32k. These results reinforce BMAttn's competitive position, both in terms of latency and throughput, and highlight its strength in real-world applications that require large-context attention.
>
>
>
> | Method     | 1K        | 2K        | 4K         | 8K         | 16K        | 32K         |
> | ---------- | --------- | --------- | ---------- | ---------- | ---------- | ----------- |
> | FA2        | 3.091     | 6.390     | 13.341     | 27.881     | 63.176     | 151.772     |
> | SA-8b      | 3.058     | 6.265     | 12.872     | 26.148     | 56.143     | 124.496     |
> | SA2-4b     | 3.036    | 6.221     | 12.738     | 25.538     | 53.718     | 115.556     |
> | SpargeAttn | 4.729     | 7.604     | 14.433     | 26.943     | 55.627     | 116.481     |
> | BMAttn     | **3.033** | **6.210** | **12.699** | **25.413** | **53.268** | **113.949** |

---

> > ### Comment · Reviewer_9Diq · 2025-11-25
> > **Reply For Rebuttal**
> >
> > Given the new additions in Appendix F, the expanded comparison with Sparge, and the authors’ thorough responses, I am raising my score to weak accept. The microbenchmarks are an important step toward demonstrating real speedup; however, the definitive evaluation requires TPOT, TTFT, and full end-to-end latency. Including these metrics would further strengthen the credibility of the work.

---

> > > ### Author Response · Authors · 2025-11-28
> > >
> > > Thank you for your thoughtful comments and for raising your score. Regarding your suggestion to provide TPOT, TTFT, and full end-to-end latency metrics: as mentioned in our response to Weakness #2, we have added real TTFT measurements collected on an NVIDIA RTX 4090, and these results have been included in the revised manuscript in Section E.4.
> > >
> > > To further strengthen the practical evidence, we additionally conducted speed benchmark of Qwen2-7b and Llama3.1-8b on NVIDIA L20 GPUs, as shown in the table below. These experiments demonstrate that BMAttn achieves significant improvements in TTFT and end-to-end serving latency compared to the latest Sage+SpargeAttn kernels, further validating the efficiency and deployability of our approach.
> > > | Method                                 | Benchmark Durations (s) | TTFT (ms) | TPOT (ms) | ITL (ms) |
> > > | -------------------------------------- | :---------------------: | --------- | --------- | -------- |
> > > | **Qwen2-7b**                           |          16.62          | 195.09    | 21.69     | 21.47    |
> > > | +SageAttn2++_8bit+SpargeAttn(topk=0.5) |          14.78          | 150.06    | 21.65     | 21.50    |
> > > | +BMAttn                                |          14.49          | 146.68    | 21.66     | 21.48    |
> > > | **Llama3.1-8b**                        |          17.52          | 203.28    | 22.90     | 22.63    |
> > > | +SageAttn2++_8bit+SpargeAttn(topk=0.5) |          15.47          | 156.36    | 22.92     | 22.71    |
> > > | +BMAttn                                |          15.07          | 152.84    | 22.88     | 22.65    |
> > >
> > >
> > > As with SageAttn and SpargeAttn, BMAttn primarily targets the prefill phase. Therefore, during the decode phase, TPOT and ITL remain comparable to the baseline FlashAttention implementation. Nonetheless, the strong end-to-end performance gains observed on real deployment hardware (NVIDIA L20 and RTX4090) demonstrate that BMAttn provides substantial real-world benefits beyond microbenchmark-level speedups.
> > >
> > > Thank you again for the constructive feedback and positive evaluation.

---

### Meta-Review · Area_Chair_7xqM · 2025-12-09

**Summary:**

This paper introduces Block-Aligned Mixed-Precision Attention (BMAttn), a framework that accelerates LLM inference by partitioning the attention map into three distinct distance-based precision zones (high precision, low precision, and sparse). Though the theoretical guarantee essentially notes that inverse-propensity weighting results in an unbiased estimator, the empirical evidence achieves up to 3x speed-up while retaining good across three modern LLM families (MMLU for multi-task understanding, LongBench for long-context reasoning, and RULER for reasoning and reading comprehension). However, reviewers express concerns about two main issues: (1) limited conceptual novelty, as the method is based on known techniques (quantization, pruning, windowing); and (2) an initial sparsity of kernel-level evidence, e.g., profiling data, pseudocode, memory bandwidth traces, to demonstrate that the kernel avoids low-level hardware inefficiencies to support the claims of the paper.

A substantial number of subsequent experiments were performed during the rebuttal phase and integrated into a revised version. While these additional experiments resolve a number of questions about experiments, concerns about the number of benchmarks remain, given the somewhat limited conceptual novelty of the paper.

**Reviewer Concerns:**

I believe concerns about comparisons to other baselines such as SpargeAttn should have been partially or even fully addressed. Some concerns about kernel-level details may have been partially addressed as well. However, I believe concerns about the number of benchmarks (datasets) and the conceptual novelty may still be outstanding.

**Reviewer Scores:**

Reviewer 9Diq increased their score from 2 to 6.
I believe concerns of Reviewer K21C and Reviewer QYg9 about other baselines may have been partially addressed, but concerns about conceptual novelty were not comprehensively addressed, so I do not believe they would have substantially increased their score.
Similarly, I believe some concerns of Reviewer hcmZ and Reviewer QYg9 regarding kernel profiling may have been partially addressed, but not necessarily to the desired extent.
Furthermore, I agree that these concerns are valid, as given the concerns about the algorithmic novelties of the methods, the practical improvements of the paper should be thoroughly clear.

---

### Decision · Program_Chairs · 2026-01-26

Reject